# PRES: Toward Scalable Memory-Based Dynamic Graph Neural Networks

**Junwei Su, Difan Zou**[†] **& Chuan Wu**[†]
Department of Computer Science, University of Hong Kong
{jwsu,dzou,cwu}@cs.hku.hk

## ABSTRACT

Memory-based Dynamic Graph Neural Networks (MDGNNs) are a family of dynamic graph neural networks that leverage a memory module to extract, distill, and memorize long-term temporal dependencies, leading to superior performance compared to memory-less counterparts. However, training MDGNNs faces the challenge of handling entangled temporal and structural dependencies, requiring sequential and chronological processing of data sequences to capture accurate temporal patterns. During the batch training, the temporal data points within the same batch will be processed in parallel, while their temporal dependencies are neglected. This issue is referred to as temporal discontinuity and restricts the effective temporal batch size, limiting data parallelism and reducing MDGNNs' flexibility in industrial applications. This paper studies the efficient training of MDGNNs at scale, focusing on the temporal discontinuity in training MDGNNs with large temporal batch sizes. We first conduct a theoretical study on the impact of temporal batch size on the convergence of MDGNN training. Based on the analysis, we propose PRES, an iterative prediction-correction scheme combined with a memory coherence learning objective to mitigate the effect of temporal discontinuity, enabling MDGNNs to be trained with significantly larger temporal batches without sacrificing generalization performance. Experimental results demonstrate that our approach enables up to a $4 \times$ larger temporal batch ($3.4\times$ speed-up) during MDGNN training.

## 1 INTRODUCTION

Graph representation learning has become increasingly important due to its ability to leverage both feature vectors and relational information among entities, providing powerful solutions in various domains (Wu et al., 2020; Hamilton, 2020). While initial research (Kipf & Welling, 2017; Hamilton et al., 2018; Veličković et al., 2017) primarily focuses on *static graphs*, many real-world applications involve *dynamic graphs* (also referred to as *temporal graph*) characterized by continuously changing relationships, nodes, and attributes. To address this dynamic nature, dynamic graph neural networks (DGNNs) have emerged as promising deep learning models capable of modelling time-varying graph structures (Kazemi et al., 2020; Skarding et al., 2021). Unlike their static counterparts, DGNNs excel at capturing temporal dependencies and learning spatial representations within the context of dynamic graphs. Consequently, they play a critical role in applications such as social media (Zhang et al., 2021), where communication events stream and relationships evolve, and recommender systems (Kumar et al., 2019), where new products, users, and ratings constantly emerge.

Among DGNNs, Memory-based Dynamic Graph Neural Networks (MDGNNs) have demonstrated superior performance compared to memory-less counterparts (Poursafaei et al., 2022). An essential feature of MDGNNs is the integration of a memory module within their architectures. This memory module acts as a sequential filtering mechanism, iteratively learning and distilling information from both new and historical graph data (also referred to as events). Consequently, MDGNNs excel at capturing long-range dependencies and achieving state-of-the-art performance across a diverse range of dynamic-graph-related tasks (Zhang et al., 2023; Rossi et al., 2021; Xu et al., 2020). The

---

The implementation is available at: https://github.com/jwsu825/MDGNN_BS
†: corresponding authors

training and inference processes of MDGNNs involve three key steps. First, the incoming events are sequentially processed in their temporal order using the MESSAGE module, which extracts and distils relevant information, generating a set of message vectors. These message vectors are then employed, in conjunction with the previous memory state, by the MEMORY module to update the memory states of the vertices involved in the events. Finally, the updated memory vectors, along with structural information, are fed into the EMBEDDING module to generate dynamic embeddings for the vertices. Fig. 1 visually depicts this process, illustrating the information flow within MDGNNs.

Because of its effectiveness, there is a recent uptick in both theoretical exploration (expressive power) (Souza et al., 2022) architectural innovation (Zhang et al., 2023) concerning MDGNN. However, the presence of the memory module poses a challenge for MDGNN training. Capturing accurate temporal patterns requires sequential and chronological processing of event sequences (Rossi et al., 2021; Zhou et al., 2022). To achieve training efficiency, MDGNNs adopt batch processing, where consecutive events are partitioned into *temporal batches* and each batch of events is fed to the model concurrently for processing. However, this can be problematic as the temporal dependency between the data points within the same batch cannot be maintained. When events involving the same node coexist within a batch, batch processing only results in one update for

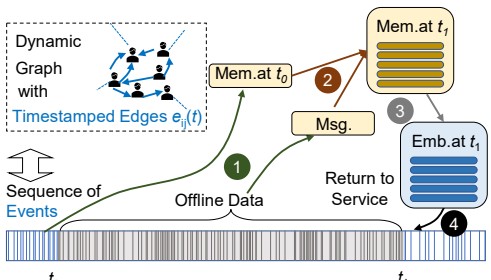

Figure 1: Illustration of the MDGNN process. Arrows of the same colour represent simultaneous operations. (1) Temporal events are sequentially processed and transformed into messages. (2) Arrived messages update the previous memory. (3) The updated memory is used to compute embeddings. (4) Time-dependent embeddings can be utilized for downstream tasks within the system.

MDGNNs, whereas principally, the MESSAGE and MEMORY should be updated twice based on the chronological order of the events (see Sec. 3.1 for a detailed discussion and Fig. 2(b) for a visual illustration). This phenomenon is known as the *temporal discontinuity* which can lead to information loss and performance deterioration. Clearly, this issue will become more severe when using a large batch size, as a greater number of events (which have temporal order) will be included in the same batch and processed simultaneously. Therefore, in practice, one may need to use a small temporal batch size in training MDGNNs (Rossi et al., 2021; Kumar et al., 2019; Zhou et al., 2022), restricting data parallelism and reducing flexibility in practical applications. In contrast, modern deep learning methods have achieved remarkable success by training models on large amounts of data, leveraging large batch sizes to exploit the computational power of parallel processing (Goyal et al., 2017; You et al., 2019; 2020; Rasley et al., 2020). Addressing the batch size bottleneck is crucial to enhance the practicality of MDGNNs, enabling more efficient training, optimized utilization of computation resources, and improved performance and applicability across diverse domains.

This paper aims to enhance the efficiency of MDGNN training by investigating the impact of temporal batch size on the training procedure. Our contributions are summarized as follows:

• We provide a formal formulation of the MDGNN training procedure and present, to the best of our knowledge, the first theoretical result on the influence of temporal batch size on MDGNN training. Contrary to the prevailing belief that smaller temporal batches always yield better MDGNN performance, we demonstrate that small temporal batches can introduce significant variance in the gradient (Theorem 1). Furthermore, leveraging the concept of memory coherence (Def. 3), we present a novel convergence result (Theorem 2) for MDGNN training, offering insights into the factors impacting the convergence rate.

• Building upon the aforementioned analysis and drawing inspiration from the similarity between the memory module and the filtering mechanism in control theory, we propose a novel training framework for MDGNNs, named PRES (PREdict-to-Smooth), which offers provable advantages (Proposition 1). PRES consists of two key components: 1) an iterative prediction-to-correction scheme that can mitigate the variance induced by temporal discontinuity in large temporal batches, and 2) a memory smoothing learning objective aiming at improving the convergence rate of MDGNNs.

• To validate our theoretical analysis and the effectiveness of our proposed method, we conduct an extensive experimental study. The experimental results (Sec. 6.1) demonstrate that our approach

enables the utilization of up to $4 \times$ larger temporal batch ($3.4\times$ speed-up) during MDGNN training, without compromising model generalization performance.

## 2 RELATED WORK

Due to space limit, here we discuss papers that are most relevant to the problem we study and provide a more comprehensive review in Appendix G.

**Dynamic Graph Representation Learning.** Dynamic graph representation learning has garnered substantial attention in recent years, driven by the imperative to model and analyze evolving relationships and temporal dependencies within dynamic graphs (Skarding et al., 2021; Kazemi et al., 2020). Dynamic Graph Neural Networks (DGNNs), as dynamic counterparts to GNNs, have emerged as promising neural models for dynamic graph representation learning (Sankar et al., 2020; Poursafaei et al., 2022; Xu et al., 2020; Rossi et al., 2021; Wang et al., 2021; Kumar et al., 2019; Trivedi et al., 2019; Zhang et al., 2023; Pareja et al., 2020; Trivedi et al., 2017). Among DGNNs, MDGNNs have demonstrated superior inference performance compared to their memory-less counterparts. Consequently, there has been a recent surge in both theoretical exploration (expressive power)(Souza et al., 2022) and architectural innovation(Rossi et al., 2021; Wang et al., 2021; Kumar et al., 2019; Trivedi et al., 2019; Zhang et al., 2023) related to MDGNNs. Additionally, there are works dedicated to optimizing both the inference and training efficiency of MDGNNs from a system perspective, employing techniques such as computation duplication (Wang & Mendis, 2023), CPU-GPU communication optimization (Zhou et al., 2022), staleness (Sheng et al., 2024) and caching (Wang et al., 2021). Despite the recognition of the temporal discontinuity problem (which may be referred to differently) (Zhou et al., 2022; Rossi et al., 2021; Kumar et al., 2019), there are still no theoretical insights or founded solutions addressing the temporal discontinuity issue (limited temporal batch size). Our study addresses this gap in MDGNN research by focusing on understanding the impact of temporal batch size on training MDGNNs, aiming to enhance data parallelism in MDGNN training. We adopt a more theoretical approach, and our proposed method can be used in conjunction with the aforementioned prior works to further improve training scalability

**Mini-Batch in Stochastic Gradient Descent (SGD).** It should be noted that there is another orthogonal line of research investigating the effect of mini-batch size in SGD training (Goyal et al., 2017; Qian & Klabjan, 2020; Lin et al., 2018; Akiba et al., 2017; Gower et al., 2019), including studies in the context of GNNs (Chen et al., 2018; 2017; Ying et al., 2018; Huang et al., 2018). It is important to differentiate the concepts of mini-batch in SGD and temporal batch in MDGNNs, as they serve distinct purposes and bring different challenges. The goal of mini-batches in SGD is to obtain a good estimation of the full-batch gradient by downsampling the entire dataset into mini-batches. On the other hand, the temporal batch specifically refers to partitioning consecutive events data to ensure chronological processing of events. The temporal batch problem studied in this paper aims to increase the temporal batch size to enhance data parallelism in MDGNN training.

## 3 PRELIMINARY AND BACKGROUND

**Event-based Representation of Dynamic Graphs.** In this paper, we utilize event-based representation of dynamic graphs, as described in previous works (Skarding et al., 2021; Zhang et al., 2023). A dynamic graph $\mathcal{G}$ in this representation consists of a node set $\mathcal{V} = \{1, ..., N\}$ and an event set $\mathcal{E} = \{e_{ij}(t)\}$, where $i, j \in \mathcal{V}$. The event set $\mathcal{E}$ represents a stream of events, where each edge $e_{ij}(t)$ corresponds to an interaction event between node $i$ and node $j$ at timestamp $t \geq 0$. Node features and edge features are denoted by $v_i(t)$ and $e_{ij}(t)$, respectively. In the case of non-attributed graphs, we assume $v_i(t) = 0$ and $e_{ij}(t) = 0$, representing zero vectors.

**Memory-based Dynamic Graph Neural Network (MDGNN).** We adopt an encoder-decoder formulation of MDGNNs, following the setting in (Rossi et al., 2021; Zhang et al., 2023). The encoder takes a dynamic graph as input and generates dynamic representations of nodes, while the decoder utilizes these node representations for various downstream tasks. Given event $e_{ij}(t)$, the

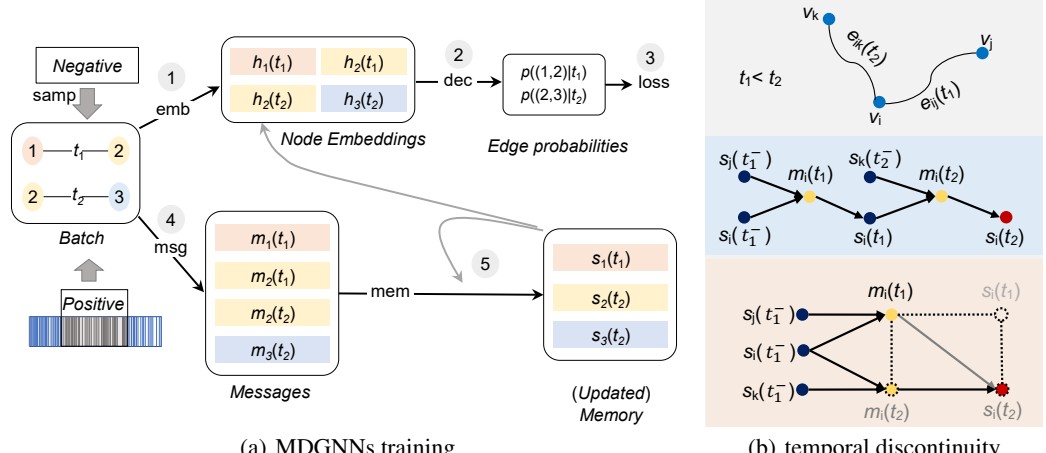

Figure 2: Fig. 2(a) depicts the training flow of MDGNN. The incoming batch serves as training samples for updating the model and memory for the subsequent batch. Fig. 2(b) visualizes the temporal discontinuity that arises from pending events within the same temporal batch and $t^-$ indicates the moments before $t$. The top section showcases two pending events sharing a common vertex. The middle section demonstrates the transition of memory states when events are sequentially processed according to temporal order. The bottom section illustrates the transition when events are processed in parallel (large batch). The grey colour indicates unobserved or altered memory states and the dotted line indicates missing transition, resulting in temporal discontinuity.

encoder of MDGNN can be defined as follows:

$$m_i(t) = \text{msg}(s_i(t^-), s_j(t^-), e_{ij}(t), \Delta t), \quad m_j(t) = \text{msg}(s_j(t^-), s_j(t^-), e_{ij}(t), \Delta t),$$
$$s_i(t) = \text{mem}(s_i(t^-), m_i(t)), \quad s_j(t) = \text{mem}(s_j(t^-), m_j(t)), \tag{1}$$
$$h_i(t) = \text{emb}(s_i(t), \mathcal{N}_i(t)), \quad h_j(t) = \text{emb}(s_j(t), \mathcal{N}_j(t)),$$

where $s_i(t^-)$ and $s_j(t^-)$ are the memory states of nodes $i$ and $j$ just before time $t$ (i.e., at the time of the previous interaction involving node $i$ or $j$), $m_i(t)$ and $m_j(t)$ are the messages generated from the event $e_{ij}(t)$, $\mathcal{N}_i(t)$ and $\mathcal{N}_j(t)$ are the temporal neighbours of nodes $i$ and $j$ up to time $t$, $h_i(t)$ and $h_j(t)$ are the dynamic embeddings of nodes $i$ and $j$ at time $t$, and $\text{msg}(.)$ (e.g., MLP), $\text{mem}(.)$(e.g., GRU), and $\text{emb}(.)$(e.g., GCN) are learnable functions, respectively, representing the MESSAGE, MEMORY, and EMBEDDING modules discussed earlier.

**Training MDGNNs.** While MDGNNs possess the theoretical capability of capturing continuous dynamics, their training relies on batch processing to account for data discretization and efficiency considerations (Rossi et al., 2021; Zhang et al., 2023). In the training process of MDGNN on a dynamic graph with an event set $\mathcal{E}$, consecutive events are partitioned into temporal batches $B_1, \ldots, B_K$ of size $b = \frac{|\mathcal{E}|}{K}$. These temporal batches are sequentially trained to capture the correct temporal relationships. MDGNNs are commonly trained using the (self-supervised) temporal link prediction task (Rossi et al., 2021; Zhang et al., 2023; Xu et al., 2020), which involves binary classification to predict whether an interaction will occur between a given pair of vertices. The event set $\mathcal{E}$ provides positive signals (interactions) for the temporal link prediction task. For each batch $B_i$, the negative signals are formed by considering vertex pairs that do not have an event within the time interval of the batch (Xu et al., 2020; Rossi et al., 2021; Zhang et al., 2023), denoted as $\bar{B}_i$. The complete event batch used for training is represented as $\mathcal{B}_i = B_i \cup \bar{B}_i$. To prevent information leakage of $\mathcal{B}_i$ (can not predict $\mathcal{B}_i$ with its own information), MDGNNs adopt a lag-one scheme (Rossi et al., 2021; Zhang et al., 2023) where the temporal batch $\mathcal{B}_{i-1}$ is used to update the memory state and generate embeddings for predicting $\mathcal{B}_i$. The training process in each training epoch with gradient descent can be formulated as the following iterative process:

$$\mathcal{L}(\theta^{(0)}) := \sum_{i=1}^{K} \mathcal{L}_i(\theta^{(i-1)}), \quad \mathcal{L}_i(\theta^{(i-1)}) = l(\mathcal{B}_i, \theta^{(i-1)}), \quad \theta^{(i)} = \theta^{(i-1)} - \eta \nabla \mathcal{L}_i(\theta^{(i-1)}) \tag{2}$$

where $\theta$ represents the model parameters of the MDGNN, $\eta$ is the learning rate, $\mathcal{L}(\theta^{(0)})$ is the total loss for the entire epoch with initial parameters $\theta^{(0)}$, $\mathcal{L}_i(.)$ is the loss for batch $\mathcal{B}_i$, and $l(.)$

denotes the loss function (e.g., cross-entropy (Zhang et al., 2023; Xu et al., 2020; Rossi et al., 2021)). Interactions in real-life networks are usually sparse, and it is impractical to process the complete negative event set. Instead, a subset of negative events is sampled to facilitate training. We denote the temporal batch with negative sampling as $\hat{\mathcal{B}}_i$ and its gradient as $\nabla\hat{\mathcal{L}}_i(\theta^{(i-1)})$ (can be obtained by replacing $\mathcal{B}_i$ in equation 2), and the training updates become,

$$\theta^{(i)} = \theta^{(i-1)} - \eta\nabla\hat{\mathcal{L}}_i(\theta^{(i-1)}). \tag{3}$$

We denote $\nabla\hat{\mathcal{L}}(\theta)$ as the gradient of the entire epoch with negative sampling, and make the following assumption on the sampling process of negative events:

**Assumption 1.** $\nabla\hat{\mathcal{L}}_i(\theta^{(i-1)})$ *is an unbiased estimate of* $\nabla\mathcal{L}_i(\theta^{(i-1)})$ *and has bounded variance, i.e.,* $\sigma_{\min}^2 \leq \mathbb{E}[\|\nabla\hat{\mathcal{L}}_i(\theta^{(i-1)}) - \nabla\mathcal{L}_i(\theta^{(i-1)})\|^2] \leq \sigma_{\max}^2$.

This assumption ensures that the variance of the gradient estimation remains bounded during the sampling process. Fig. 2(a) provides a graphical illustration of MDGNN training procedure. Appendix A contains pseudocode for the training procedure, along with a more detailed description.

### 3.1 TEMPORAL DISCONTINUITY AND PENDING EVENTS.

Training MDGNNs using temporal batches introduces challenges in accurately capturing temporal patterns. Specifically, events involving the same vertex within a batch are inherently temporal and should be processed chronologically to ensure the memory module accurately captures the temporal patterns, as depicted in the central portion of Fig.2(b). Batch processing, however, only triggers one update for the MDGNNs, neglecting the temporal dependency of events, as illustrated in the lower section of Fig.2(b). This phenomenon is termed temporal discontinuity and can lead to information loss and potential noise in the memory state. To formally define this concept, we introduce the terms pending events and pending sets as follows:

**Definition 1** (Pending Event). *An event $e$ is considered pending on another event $e'$ (denoted as $e' \rightarrow e$) if $e$ and $e'$ share at least one common vertex and $t' < t$, where $t$ ($t'$) is the timestamp of event $e$ ($e'$).*

**Definition 2** (Pending Set). *Given a temporal batch $B$ and an event $e$, the pending set $\mathcal{P}(e, B) = \{e' \rightarrow e | e' \in B\}$ of an event $e$ is the collection of its pending events within batch $B$.*

In essence, as the temporal batch size increases, it tends to accommodate a greater number of pending events within the batch. As a result, training MDGNNs with large temporal batches becomes a formidable task since concurrently processing numerous pending events can result in the loss of critical temporal information within the graph. This limitation imposes constraints on the effective temporal batch size, thereby impeding training parallelization.

## 4 THEORETICAL ANALYSIS OF MDGNN TRAINING

In this section, we provide a theoretical analysis of MDGNN training, specifically focusing on how the size of the temporal batch affects the variance and convergent rate of the learning process.

**Theorem 1.** *Let $\mathcal{E}$ be a given event set and $b$ be the temporal batch size. For a given MDGNN parameterized by $\theta$ with training procedure of Eq. 3, we have $\mathbb{E}[\|\nabla\hat{\mathcal{L}}(\theta) - \nabla\mathcal{L}(\theta)\|^2] \geq \frac{|\mathcal{E}|}{b}\sigma_{\min}^2$.*

The proof of Theorem 1 can be found in Appendix B. Theorem 1 provides an interesting insight into MDGNN training: while it is commonly believed that a smaller temporal batch size leads to better information granularity (less subject to pending events), the variance of the gradient for the entire epoch can be significant when the temporal batch size is small. This underscores the unexpected benefit of larger temporal batch sizes, as they exhibit greater robustness to sampling noise.

Next, we examine how the size of the temporal batch affects the convergence of the training process. We introduce the concept of memory coherence to measure the loss of temporal dependency in learning dynamics. Specifically, let $s_i^{(e)}$ denote the memory state of vertex $i$ after event $e$ (i.e., $s_i^{(e)} = s_i(t_e)$), and we define memory coherence as follows.

**Definition 3** (Memory Coherence). *The memory coherence of an event $e_{ij} \in \mathcal{B}_k$ is defined as*

$$\mu_{e_{ij}}(\mathcal{B}_k) := \min_{e \in \mathcal{P}(e_{ij}, \mathcal{B}_k)} \frac{\left\langle \nabla l(e_{ij}, s_i^{(e)}, s_j^{(e)}), \nabla l(e_{ij}, s_i^{(e_{ij})}, s_j^{(e_{ij})}) \right\rangle}{\|\nabla l(e_{ij}, s_i^{(e_{ij})}, s_j^{(e_{ij})})\|^2} \tag{4}$$

where $\nabla l(e_{ij}, s_i^{(e)}, s_j^{(e)})$ is the gradient incurred by event $e_{ij}$ with respect to $s_i^{(e)}$ and $s_j^{(e)}$ (similarly for $\nabla l(e_{ij}, s_i^{(e_{ij})}, s_j^{(e_{ij})})$). The memory coherence $\mu_{e_{ij}}$ captures the minimum coherence of the gradients when fresh memory ($s_i^{(e_{ij})}$) and past memory from the pending events ($s_i^{(e)}$) are used. Intuitively, a positive value of $\mu_{e_{ij}}$ indicates that the directions of these gradients are well aligned. In other words, the convergence properties are less affected by the pending events. It is worth noting that the memory coherence $\mu_{e_{ij}}$ can be easily computed empirically during the training process.

**Theorem 2.** *Let $\mathcal{E}$ be a given event set, $b$ be the temporal batch size, and $K = \frac{|\mathcal{E}|}{b}$ be the number of temporal batches. Let $\nabla \mathcal{L}(\theta_t)$ be the gradient of epoch $t$ when all the events are processed in sequential order (no temporal discontinuity), and $\nabla \hat{\mathcal{L}}(\theta_t)$ be the gradient when events in temporal batches are processed in parallel. Suppose the training algorithm is given by,*

$$\theta_t^{(i+1)} = \theta_t^{(i)} - \eta_t \nabla \hat{\mathcal{L}}_{i+1}(\theta_t^{(i)}), \quad \theta_{t+1} = \theta_{t+1}^{(0)} = \theta_t^{(K)}, \tag{5}$$

*where $\nabla \hat{\mathcal{L}}_{i+1}(\theta_t^{(i)})$ is the gradient of batch $\hat{\mathcal{B}}_{i+1}$ when events are processed in parallel and $\eta_t$ is the learning rate. Suppose (1) $\sigma$ be as given in Assumption 1; (2) the loss function is continuously differentiable and bounded below, and the gradient is L-Lipschitz continuous; (3) the memory coherence is lower bounded by some constant $\mu$. Choosing step size $\eta_t = \frac{\mu}{L\sqrt{Kt}}$, we have*

$$\min_{0 \leq t \leq T} \mathbb{E}[\|\nabla \mathcal{L}(\theta_t)\|^2] \leq \left( \frac{2\sqrt{K}L(\mathcal{L}(\theta_0) - \mathcal{L}(\theta^*))}{\mu^2} + \sqrt{K}\sigma_{\max}^2 \log T \right) \frac{1}{\sqrt{T}} \tag{6}$$

*where $\theta^*$ is the optimal parameter and $\theta_0$ is the initial parameter.*

The proof of Theorem 2 can be found in Appendix C. The assumptions regarding the loss function, as presented in the theorem, adhere to the standard conventions in convergence analysis (Wan et al., 2022; Chen et al., 2022; Dai et al., 2018). These assumptions can be relaxed with more specific insights into the neural architecture employed in MDGNNs

The insights provided by Theorem 2 offer valuable guidance for the training process of MDGNNs. Firstly, it emphasizes the consideration that the choice of the step size $\eta_t$ should account for both the temporal batch size (as captured by $K$), and the memory coherence $\mu$. Secondly, the theorem sheds light on the pivotal roles played by memory coherence $\mu$ and variance $\sigma_{\max}$, as captured by Eq. 6, in the convergence behaviour of MDGNNs. Specifically, it underscores the significance of enhancing memory coherence $\mu$ while simultaneously reducing $\sigma_{\max}$. These insights provide strong motivation for the development of our proposed method, as elaborated below.

## 5 PREDICT-TO-SMOOTH (PRES) METHOD

We introduce PRES, our proposed method to improve MDGNN training with large temporal batch sizes. PRES is based on the theoretical analysis presented in the previous section. It consists of two main components: 1) an iterative prediction-correction scheme that mitigate the impact of temporal discontinuity in training MDGNNs (optimizing the second term of Eq. 6), and 2) a memory smoothing objective based on memory coherence (optimizing the first term of Eq. 6). An overview of PRES is given in Fig. A.2. The pseudo-code for the complete procedure is provided in Appendix A.

### 5.1 ITERATIVE PREDICTION-CORRECTION SCHEME

To tackle temporal discontinuity, we employ an iterative prediction-correction scheme inspired by filtering mechanisms used in control systems to adjust noisy measurements. The scheme consists of two steps: 1) prediction and 2) update. In the prediction step, a prediction model is used to estimate the current state of the system based on the previous state information. In the update step,

the estimation from the prediction model is employed to incorporate new measurement information and improve the state estimation (reducing the noise to the true value). Here, we treat the memory state with pending events within the temporal batch as a noisy measurement.

Our prediction model is tasked with modelling the change in the memory state $s_i$ for each vertex $i$, represented as $\delta_{s_i}$. We employ a Gaussian Mixture Model (GMM), $\mathbb{P}(\delta_{s_i}) = \sum_{j=1}^{\omega} \alpha_j \mathbb{N}(\delta_{s_i} | \mu_i^{(j)}, \Sigma_i^{(j)})$ as the parametric model to estimate the distribution of $\delta_{s_i}$. $\omega$ represents the number of components in GMM, $\alpha_j$ denotes the weights of each components, and $\mu_i^{(j)}$ and $\Sigma_i^{(j)}$ are the mean and covariance matrices for the components. We set $\omega = 2$ to model the positive and negative event types in temporal link prediction. Using this GMM, we predict the newest memory state $s_i(t_2)$ of vertex $i$ based on its previous memory state $s_i(t_1)$ and the estimated transition:

$$\hat{s}_i(t_2) = s_i(t_1) + (t_2 - t_1)\delta_{s_i}. \tag{7}$$

To make use of the predicted value, we consider the memory state as a noisy measurement affected by temporal discontinuity. We then fuse the predicted value and the noisy measurement to obtain a better estimate of the memory state. The correction step is given by:

$$\bar{s}(t_2) = (1 - \gamma)\hat{s}_i(t_2) + \gamma s_i(t_2), \tag{8}$$

where $\gamma$ is a learnable variable controlling the fusion of the predicted and the current memory state.

Then, we adopt the common approach, Maximum Likelihood Estimation (MLE), for estimating and updating the parameters of GMM. Nevertheless, applying MLE naively would require storing the complete history of each vertex, resulting in substantial memory overhead. To address this, we utilize the variance formula $\mathrm{Var}(X) = E[X^2] - E[X]^2$, which allows us to keep track of only a few key values for parameter updates. For each event of type $j$, the parameter updates are calculated as follows:

$$\delta_{s_i}^{(j)} = \bar{s}(t_2) - \hat{s}_i(t_2), \quad \xi_i^{(j)} = \xi_i^{(j)} + \delta_{s_i}^{(j)}, \quad \psi_i^{(j)} = \psi_i^{(j)} + (\delta_{s_i}^{(j)})^2,$$
$$n_i^{(j)} = n_i^{(j)} + 1, \quad \mu_i^{(j)} = \xi_i^{(j)} / n_i^{(j)}, \quad \Sigma_i^{(j)} = \psi_i^{(j)} / n_i^{(j)} - (\mu_i^{(j)})^2, \tag{9}$$

where $n_i^{(j)}, \xi_i^{(j)}, \psi_i^{(j)}$ are trackers for the number of events, the sum of $\delta_{s_i}^{(j)}$ and the sum of $(\delta_{s_i}^{(j)})^2$.

To summarize, the iterative prediction-correction scheme perceives the memory state impacted by temporal discontinuity as a noisy measurement. It then aims to utilize historical information of the memory state and a GMM prediction model to mitigate this noise/variance, akin to the filtering techniques employed in control systems.

## 5.2 MEMORY COHERENCE SMOOTHING

Based on the preceding analysis, we have identified memory coherence ($\mu$) as a pivotal determinant in the convergence rate of MDGNNs when dealing with large temporal batches. To enhance the statistical efficiency of MDGNN training with larger temporal batches, we propose a novel learning objective aimed at promoting larger memory coherence. This learning objective is expressed as:

$$\underbrace{l(\mathcal{B}_i)}_{\text{prediction loss}} + \beta \left[ 1 - \underbrace{\left\langle \frac{S^{(-)}(\mathcal{B}_i)}{\|S^{(-)}(\mathcal{B}_i)\|}, \frac{S(\mathcal{B}_i)}{\|S(\mathcal{B}_i)\|} \right\rangle}_{\text{memory coherence}} \right], \tag{10}$$

where $S^{(-)}(\mathcal{B}_i)$ and $S(\mathcal{B}_i)$ represent the previous and new memory states of vertices within batch $\mathcal{B}_i$, respectively. The hyperparameter $\beta$ governs the strength of the regularization effect.

Intuitively, if the memory coherence within batch $\mathcal{B}_i$ is low, Eq. 10 incurs a substantial loss. Consequently, this learning objective steers the training process towards parameter values less susceptible to the influence of pending events, effectively enhancing memory coherence. In accordance with Theorem 2, this enhancement is expected to yield superior statistical efficiency.

## 5.3 THEORETICAL DISCUSSION OF PRES

**Variance.** Let STANDARD represent the standard temporal batch training with Eq. 5, and PRES denote our proposed framework. We have the following theoretical guarantee for variance reduction.

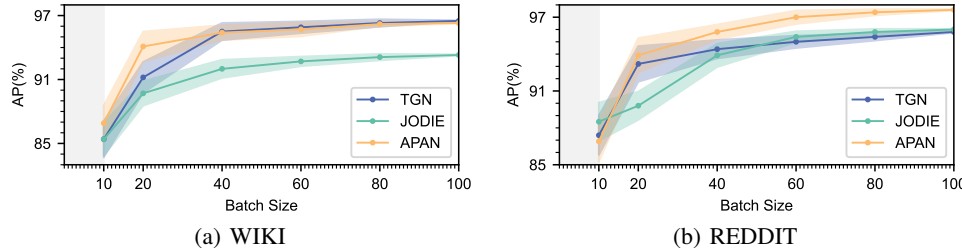

Figure 3: Performance of baselines under different batch sizes. The x-axis represents the batch size, while the y-axis represents the average precision (AP). The results are averaged over five trials.

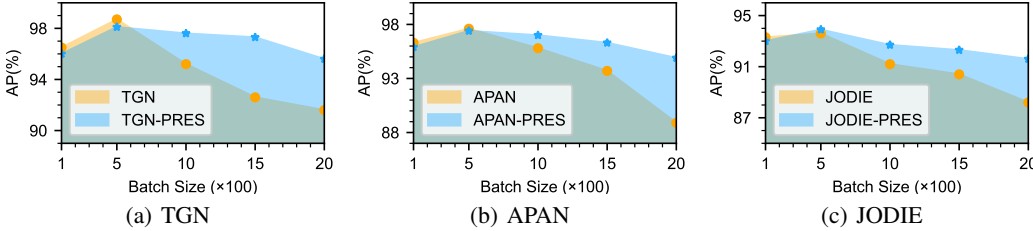

Figure 4: Performance of baseline methods with and without PRES under different batch sizes on WIKI dataset. The x-axis represents the batch size (multiplied by 100), while the y-axis represents the average precision (AP). The results are averaged over five trials with $\beta = 0.1$ for PRES.

Table 1: Performance comparison of existing MDGNNs with and without PRES. The results are averaged over five independent trials with $\beta = 0.1$ for PRES and 50 epoches. The training efficiency improvement with PRES is highlighted in bold. (*s) indicates that it takes * seconds for one epoch.

| Dataset | REDDIT | | WIKI | | MOOC | | LASTFM | | GDELT | |
|---------|--------|--------|--------|--------|--------|--------|--------|--------|--------|--------|
| Model | AP(%) | Speedup | AP(%) | Speedup | AP(%) | Speedup | AP(%) | Speedup | AP(%) | Speedup |
| TGN | $98.4 \pm 0.1$ | $1\times$ (321s) | $97.8 \pm 0.1$ | $1\times$ (87s) | $98.8 \pm 0.2$ | $1\times$ (212s) | $72.1 \pm 1.0$ | $1\times$ (648s) | $96.8 \pm 0.1$ | $1\times$(1325s) |
| TGN-PRES | $98.0 \pm 0.2$ | $\mathbf{3.4}\times$ (94s) | $97.5 \pm 0.3$ | $\mathbf{3.2}\times$ (27s) | $97.5 \pm 0.1$ | $\mathbf{3.0}\times$ (71s) | $71.2 \pm 1.5$ | $\mathbf{2.8}\times$ (231s) | $96.0 \pm 0.1$ | $\mathbf{2.7}\times$ (491s) |
| JODIE | $96.6 \pm 0.1$ | $1\times$ (271s) | $94.7 \pm 0.1$ | $1\times$ (62s) | $98.0 \pm 0.1$ | $1\times$ (152s) | $75.2 \pm 1.5$ | $1\times$ (452s) | $95.1 \pm 0.1$ | $1\times$ (1123s) |
| JODIE-PRES | $95.8 \pm 0.2$ | $\mathbf{3.1}\times$ (87s) | $94.4 \pm 0.1$ | $\mathbf{2.4}\times$ (26s) | $98.1 \pm 0.2$ | $\mathbf{2.6}\times$ (58s) | $73.2 \pm 1.6$ | $\mathbf{1.9}\times$ (237s) | $94.3 \pm 0.1$ | $\mathbf{2.8}\times$ (401s) |
| APAN | $98.6 \pm 0.1$ | $1\times$ (281s) | $99.0 \pm 0.1$ | $1\times$ (71s) | $98.5 \pm 0.1$ | $1\times$ (173s) | $69.8 \pm 1.6$ | $1\times$ (521s) | $96.7 \pm 0.2$ | $1\times$ (1215s) |
| APAN-PRES | $98.2 \pm 0.1$ | $\mathbf{2.2}\times$ (127s) | $98.5 \pm 0.1$ | $\mathbf{2.9}\times$ (24s) | $98.0 \pm 0.1$ | $\mathbf{2.0}\times$ (86s) | $67.9 \pm 1.9$ | $\mathbf{1.8}\times$ (289s) | $96.0 \pm 0.3$ | $\mathbf{2.4}\times$ (506s) |

**Proposition 1** (Informal). *If the memory transition with temporal discontinuity can be approximated by a linear state-space model with Gaussian noise,* PRES *can achieve a smaller noise/variance compared to* STANDARD.

The formal version of Proposition 1, along with its proof is provided in Appendix D. Proposition 1 provides a validation to PRES. It states PRES can effectively mitigate the variance induced by temporal discontinuity, enabling the training of MDGNN with larger temporal batches. This enhancement, in turn, bolsters the training efficiency without compromising overall performance.

**Complexity.** The computational complexity of PRES is $O(|\mathcal{B}|)$, with $|\mathcal{B}|$ representing the batch size. As for storage complexity, which arises from the trackers employed in Eq. 9, it scales at the order of $O(|\mathcal{V}|)$, where $|\mathcal{V}|$ denotes the vertex count. In cases where stringent memory constraints are in place, one can adopt a pragmatic approach by selecting a subset of vertices to serve as an anchor set. This set, in turn, facilitates an efficient heuristic for handling the remaining vertices. For a more in-depth exploration of this strategy, please refer to the appendix.

## 6 EXPERIMENT

We present an experimental study to validate our theoretical results and evaluate the effectiveness of our proposed method, PRES. Due to space limit, we present a subset of results in the main paper and provide a more comprehensive description of the datasets, experimental setup, testbed, and remaining results (such as ablation study and GPU memory utilization) in the Appendix E.

**Datasets and Baselines.** We use four public dynamic graph benchmark datasets (Kumar et al., 2019), REDDIT, WIKI, MOOC, LASTFM and GDELT. Details of these datasets are described in the appendix. We use three state-of-the-art memory-based MDGNNs models: JODIE (Kumar et al.,

2019), TGN (Rossi et al., 2021) and APAN (Wang et al., 2021), and adopt the implementation of these baselines from (Zhou et al., 2022; Rossi et al., 2021). We closely follow the settings of (Zhou et al., 2022) for hyperparameters and the chronological split of datasets, as described in more detail in the appendix. We mainly use average precision as the evaluation metric.

## 6.1 EXPERIMENTAL RESULTS

Table 2: Performance of MDGNNs w/w.o. PRES on node classification task. ROC-AUC (%)

|  | REDDIT | WIKI | MOOC |
|---|---|---|---|
| TGN | 65.6 ± 0.8 | 86.8 ± 1.5 | 57.8 ± 2.3 |
| TGN-PRES | 65.0 ± 1.2 | 84.7 ± 1.8 | 56.9 ± 3.3 |
| JODIE | 60.6 ± 3.1 | 85.2 ± 1.5 | 66.5 ± 2.0 |
| JODIE-PRES | 60.9 ± 3.5 | 85.0 ± 1.4 | 66.0 ± 1.7 |
| APAN | 64.3 ± 1.4 | 90.0 ± 1.7 | 63.5 ± 2.6 |
| APAN-PRES | 64.5 ± 1.2 | 88.6 ± 2.1 | 65.5 ± 3.6 |

**Performance vs. Batch Size.** We first examine the relationship between temporal batch size and the performance of baseline methods w./w.o. PRES. Fig. 3 provides insights into this relationship by illustrating the performance of the baselines in the small batch size regime. These results align with Theorem 1, demonstrating that contrary to common belief, smaller temporal batch sizes can lead to larger variance in the training process, and poorer performance (even non-convergence). Fig. 4 further compares the baseline methods w./w.o. PRES. The results indicate that 1) the performance of baselines indeed decreases as batch size increases and 2) baselines trained with PRES are less susceptible to performance degradation due to increased batch size, validating the effectiveness of PRES.

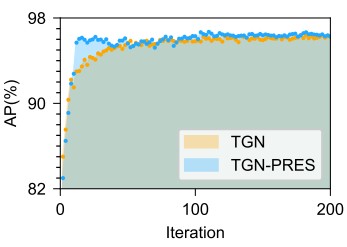

Figure 5: Statistical efficiency of baseline method w./w.o PERS. x-axis is the training iteration and y-axis is the average precision. $\beta = 0.1$ is used in PRES.

**Overall Training Efficiency Improvement.** The next experiment demonstrates the potential speed-up and efficiency improvement that baseline methods can achieve when trained with PRES. Table 1 and Table 2 provide comprehensive comparisons of the performance of baseline methods with and without PRES. The results clearly illustrate that our approach enables a significant increase in data-parallel processing by using larger temporal batch sizes, without sacrificing overall performance in terms of both link prediction and node classification tasks. These findings highlight the effectiveness of our method in significantly scaling up MDGNN training while maintaining comparable performance.

**Effectiveness of Memory Smoothing.** Next, we show the effectiveness of the memory smoothing objective. Fig. 5 demonstrates that incorporating the memory coherence-based objective indeed leads to better statistical efficiency of MDGNN training, validating the effectiveness of our approach.

## 7 CONCLUSION

This paper studies the effect of temporal batch size on MDGNN training. We present a novel theoretical analysis of how the size of the temporal batch affects the learning procedure of MDGNNs. Contrary to the common belief, we show a surprising advantage of training MDGNNs in large temporal batches (more robust to noise). In addition, we present the first convergence result of MDGNN training, illustrating the effecting factors. Based on the analysis and the filtering mechanism, we propose a novel MDGNN training framework, PRES, that can mitigate the temporal discontinuity issue and improve the convergence of MDGNN training with large temporal batch sizes. The scalability enhancements brought about by PRES hold broader implications for both researchers and practitioners, as they extend the applicability of MDGNNs to a broader spectrum of real-world scenarios featuring large-scale dynamic graphs.

**Limitations and Future Work.** In this study, our analysis centres on the MDGNN family. However, it is worth noting that the memory module has demonstrated effectiveness in capturing the dynamics of time series data, which exhibits (different) temporal dependencies. An intriguing avenue for future research would be to expand our analysis and methods to encompass the realm of time series analysis.

## ACKNOWLEDGEMENT

We would like to thank the anonymous reviewers and area chairs for their helpful comments. JS and CW are supported by grants from Hong Kong RGC under the contracts HKU 17207621, 17203522 and C7004-22G (CRF). DZ is supported by NSFC 62306252.

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

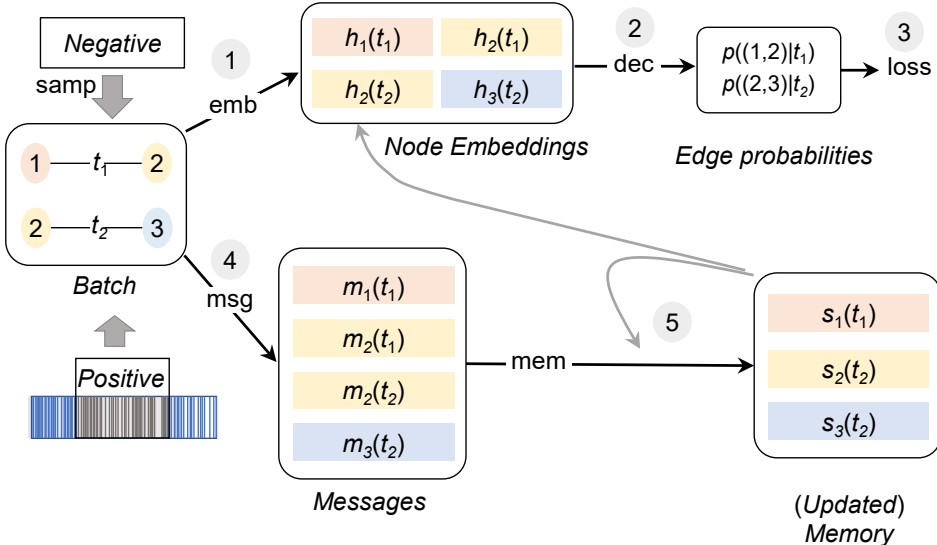

Figure 6: Illustration of MDGNN Training Procedure. Fig. A.1 depicts the training flow of MDGNN. The incoming batch serves as training samples for updating the model and updating the memory for the subsequent batch.

# A   ALGORTIHM AND FURTHER DISCUSSION

## A.1   TRAINING PROCEDURE OF MDGNNS

The training procedure of an MDGNN involves capturing temporal dynamics and learning representations of nodes in a dynamic graph. MDGNNs leverage the concept of memory to model the evolving state of nodes over time. The training process typically follows an encoder-decoder framework. In the encoder, the MDGNN takes a dynamic graph as input and generates dynamic representations of nodes. This is achieved by incorporating memory modules that store and update the past states of nodes based on their interactions and temporal neighbours. The encoder also includes message-passing mechanisms to propagate information through the graph. The decoder utilizes the node representations generated by the encoder to perform downstream tasks, such as temporal link prediction or node classification. MDGNNs are commonly trained using a self-supervised temporal link prediction task, where the decoder predicts the likelihood of an edge between two nodes based on their representations.

The training procedure of a memory-based dynamic graph neural network (MDGNN) involves several steps. First, the dataset is divided into training, validation, and test sets using a chronological split. More concretely, suppose a given event set from time interval $[0, T]$, chronological split partition the dataset into $[0, T_{\text{train}}]$ (training set), $[T_{\text{train}}, T_{\text{validation}}]$ (validation set), and $[T_{\text{validation}}, T_{\text{test}}]$ (test set). From now on, we focus on the training set and drop the subscript. The training set is then further divided into temporal batches, where each batch consists of consecutive events in the dynamic graph. In addition, negative events are sampled from the rest of the graph to provide the negative signal. During training, MDGNNs often adopt a lag-one procedure. This means that the model uses the information from the previous batch to update its memory state and generate node embeddings for the current batch. This lag-one scheme helps maintain temporal consistency and ensures that the memory-based model captures the correct temporal patterns. Fig. A.1 provides a graphical illustration of the training flow of a batch. Fig. A.1 provides a graphical illustration of the training flow between epochs. The pseudo-code of the training procedure with cross-entropy is summarized in Algorithm 2.

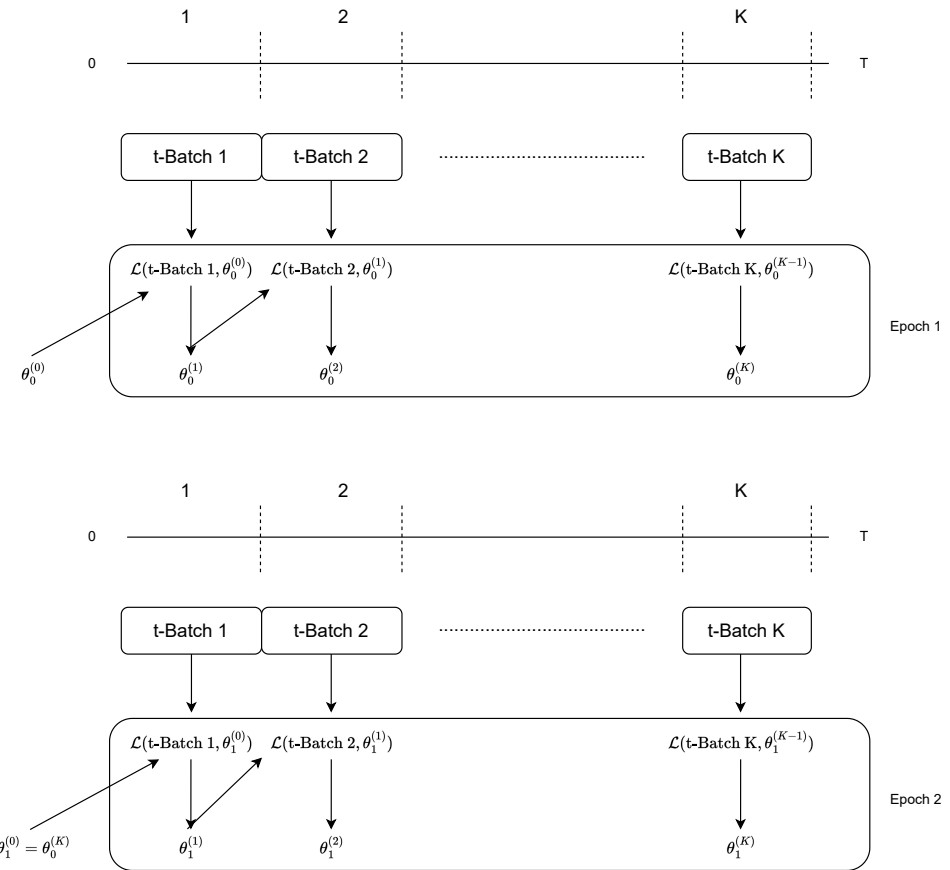

Figure 7: Illustration of MDGNN Training Procedure. Fig. A.1 depicts the training flow of MDGNN for two epoches. The incoming batch serves as training samples for updating the model and updating the memory for the subsequent batch. The model parameter is carried through the second epoch.

---

**Algorithm 1** Standard Training Procedure for Memory-based DGNN

---

**Initialization:** $S_0 \leftarrow 0$ {Initialize the memory vectors to be zero}
**for** t=1 **to** T **do**
    **for** $B_i \in B_2, ..., B_K$ **do**
        $B_i^- \leftarrow$ Sample negative events
        $\bar{B}_i = B_i^- \bigcup B_i$
        $\bar{B}_{i-1} \leftarrow$ Temporal batch from last iteration
        $M_i = \mathrm{msg}(S_{i-1}, \bar{B}_{i-1})$
        $S_i = \mathrm{mem}(S_{i-1}, M_i)$
        $H_i = \mathrm{emb}(S_i, \mathcal{N}_i)$, {where $\mathcal{N}_i$ is the (Temporal) neighbourhood of vertex }
        Compute the loss (e.g., binary cross-entropy) and run the training procedure (e.g., backpropagation)
$$\mathcal{L}(H_i, B_i)$$
    **end for**
**end for**

---

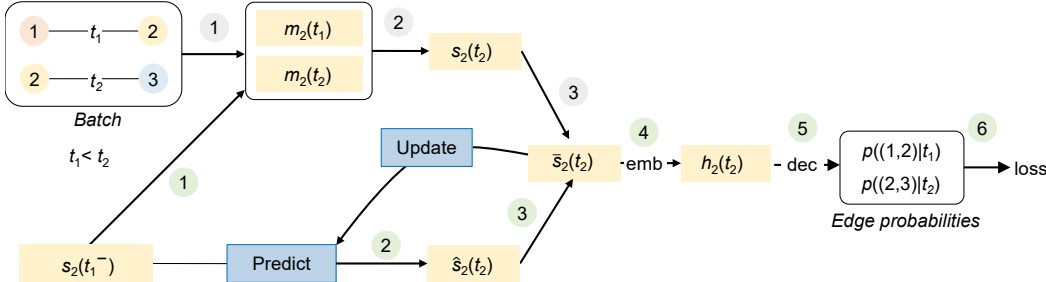

Figure 8: An illustration of MDGNN training with PRES. A prediction model is inserted into the training procedure to adjust the memory state for the intra-batch dependency. Arrows marked with the same number indicate transitions happening at the same stage.

## A.2 PRES

PRES consists of two main components: 1) an iterative prediction-correction scheme that incorporates approximation inference into training MDGNNs, and 2) a novel memory smoothing objective based on memory coherence. An overview of PRES is given in Fig. A.2. The pseudo-code for the complete procedure is summarized in Algorithm 2

---

**Algorithm 2** PRES

> **Initialization:** $S_0 \leftarrow 0$
> **for** t=1 to T **do**
> $\quad \xi, \psi, n \leftarrow \mathbf{0}$
> $\quad$ **for** $B_i \in B_1, ..., B_K$ **do**
> $\quad\quad B_i^- \leftarrow$ Sample negative events
> $\quad\quad \bar{B}_i = B_i^- \bigcup B_i$
> $\quad\quad \bar{B}_{i-1} \leftarrow$ Temporal batch from last iteration
> $\quad\quad \mathcal{S}_{\text{prev}}(\bar{B}_{i-1}) \leftarrow \mathcal{S}_{i-1}(\bar{B}_{i-1})$ {Store the previous memory state of the vertices to be updated}
> $\quad\quad M_i = \text{msg}(S_{i-1}, \bar{B}_{i-1})$
> $\quad\quad S_i = \text{mem}(S_{i-1}, M_i)$
> $\quad\quad H_i = \text{emb}(S_i, A_i)$
> $\quad\quad \hat{S}_i \leftarrow$ predict memory state with the prediction model Eq. 7.
> $\quad\quad \bar{S}_i = \gamma S_i + (1-\gamma)\hat{S}_i$
> $\quad\quad$ Compute the loss and run the training procedure (e.g., backpropagation)
>
> $$\mathcal{L}(B_i) + [1 - \langle \frac{\mathcal{S}_{\text{prev}}(\bar{B}_{i-1})}{\|\mathcal{S}_{\text{prev}}(\bar{B}_{i-1})\|}, \frac{\bar{S}_i}{\|\bar{S}_i\|} \rangle]$$
>
> $\quad\quad \delta_{S_i} = \bar{S}_i - S_i$
> $\quad\quad$ update $\xi, \psi, n$ with Eq. 9
> $\quad$ **end for**
> **end for**

---

## B  PROOF OF THEOREM 1

In this appendix, we provide a proof of Theorem 1.

*Proof.* First, recall that for a given event set of size $M$ and partition size of $\kappa$, the training procedure of MDGNN is given by

$$\mathcal{L}(\theta^{(0)}) := \sum_{i=1}^{K} \mathcal{L}_i(\theta^{(i-1)}),$$
$$\mathcal{L}_i(\theta^{(i-1)}) = l(\mathcal{B}_i, \theta^{(i-1)}) \tag{11}$$

where $K = \frac{M}{\kappa}$, $\theta$ represents the model parameters of the MDGNN, $\mathcal{L}(\theta^{(0)})$ is the total loss for the entire epoch with initial parameters $\theta^{(0)}$, $\mathcal{L}_i(.)$ is the loss for batch $\mathcal{B}_i$, and $l(.)$ denotes the loss function. Here, we consider the standard setting where negative event sampling is used to facilitate training. We denote the gradient with sampled negative events as $\nabla\hat{\mathcal{L}}_i(\theta^{(i-1)})$, and $\nabla\hat{\mathcal{L}}_i(\theta^{(i-1)})$ is used for update instead.

With the above set-up and by definition of variance, we can write $\mathrm{Var}[\nabla\mathcal{L}(\theta^{(0)})]$ as follows,

$$\mathrm{Var}[\nabla\mathcal{L}(\theta^{(0)})] = \mathbb{E}[(\sum_{i=1}^{K} \nabla\hat{\mathcal{L}}_i(\theta^{(i-1)}) - \sum_{i=1}^{K} \mathbb{E}(\nabla\hat{\mathcal{L}}_i(\theta^{(i-1)}))^2] \tag{12}$$

By Assumption 1, we assume that the negative sampling is unbiased and have that $\nabla\hat{\mathcal{L}}_i(\theta^{(i-1)})$ is an unbiased estimate of $\nabla\mathcal{L}_i(\theta^{(i)})$. Then, we can arrange the terms in the above equations and get,

$$
\begin{aligned}
\mathrm{Var}[\nabla\mathcal{L}(\theta^{(0)})] &= \mathbb{E}[(\sum_{i=1}^{K} \nabla\hat{\mathcal{L}}_i(\theta^{(i-1)}) - \sum_{i=1}^{K} \mathbb{E}(\nabla\hat{\mathcal{L}}_i(\theta^{(i-1)}))^2] \\
&= \mathbb{E}[(\sum_{i=1}^{K} \nabla\hat{\mathcal{L}}_i(\theta^{(i-1)}) - \sum_{i=1}^{K} \nabla\mathcal{L}_i(\theta^{(i-1)})^2] \\
&= \mathbb{E}[(\sum_{i=1}^{K} [\nabla\hat{\mathcal{L}}_i(\theta^{(i-1)}) - \nabla\mathcal{L}_i(\theta^{(i-1)})])^2] \\
&= (\sum_{i=1}^{K} \mathbb{E}[\nabla\hat{\mathcal{L}}_i(\theta^{(i-1)}) - \nabla\mathcal{L}_i(\theta^{(i-1)})])^2
\end{aligned} \tag{13}
$$

Again, by Assumption 1, we have that $\nabla\hat{\mathcal{L}}_i(\theta^{(i-1)})$ and $\nabla\mathcal{L}_i(\theta^{(i-1)})$ has bounded difference without lower bound of $c$. This leads to,

$$
\begin{aligned}
\mathrm{Var}[\nabla\mathcal{L}(\theta^{(0)})] &= (\sum_{i=1}^{K} \mathbb{E}[\nabla\hat{\mathcal{L}}_i(\theta^{(i-1)}) - \nabla\mathcal{L}_i(\theta^{(i-1)})])^2 \\
&\geq Kc
\end{aligned} \tag{14}
$$

Substitute in $K = \frac{M}{\kappa}$, we have

$$
\begin{aligned}
\mathbb{E}[\|\nabla\hat{\mathcal{L}}(\theta) - \nabla\mathcal{L}(\theta)\|^2] = \mathrm{Var}[\nabla\mathcal{L}(\theta^{(0)})] &\geq Kc \\
&\geq \frac{M}{\kappa}c.
\end{aligned} \tag{15}
$$

$\square$

# C   APPENDIX:PROOF OF THEOREM 2

In this appendix, we provide proof of Theorem 2. The key challenge of the proof for Theorem 2 is to keep track of the notations and the evolution of the parameters induced by batches and epochs.

Next, we provide the proof of Theorem 2.

*Proof.* Suppose $\mathcal{B}_1, ..., \mathcal{B}_K$ are the $K$ batch. Let's recall that the gradient of an epoch is given as follows,

$$\mathcal{L}(\theta^{(0)}) := \sum_{i=1}^{K} \mathcal{L}_i(\theta^{(i-1)}),$$
$$\mathcal{L}_i(\theta^{(i-1)}) = \mathcal{L}(\mathcal{B}_i, \theta^{(i-1)}). \tag{16}$$

Let's start with considering the progression within an epoch, and let $\theta_t^{(i-1)}$, denote the model parameter of epoch $t$ after being trained on batch $\mathcal{B}_{i-1}$.

We denote the gradient with sampled negative events as $\nabla \hat{\mathcal{L}}_i(\theta^{(i-1)})$, and $\nabla \hat{\mathcal{L}}_i(\theta^{(i-1)})$ is used for update instead,i.e, the update rule is given by,

$$\theta_t^{(i)} = \theta_t^{(i-1)} - \eta_t \nabla \hat{\mathcal{L}}_i(\theta^{(i-1)}), \tag{17}$$

or equivalently,

$$\theta_t^{(i)} - \theta_t^{(i-1)} = -\eta_t \nabla \hat{\mathcal{L}}_i(\theta^{(i-1)}), \tag{18}$$

By the L-Lipschitz property of the objective function $\nabla \mathcal{L}$, we have the following inequality,

$$\mathcal{L}(\theta_t^{(i+1)}) \leq \mathcal{L}(\theta_t^{(i)}) + \langle \theta_t^{(i+1)} - \theta_t^{(i)}, \nabla \mathcal{L}(\theta_t^{(i)}) \rangle + \frac{L}{2} \|\theta_t^{(i+1)} - \theta_t^{(i)}\|^2 \tag{19}$$

Substitute in $\theta_t^{(i)} - \theta_t^{(i-1)} = -\eta_t \nabla \hat{\mathcal{L}}_i(\theta^{(i-1)})$, we have that,

$$\mathcal{L}(\theta_t^{(i+1)}) \leq \mathcal{L}(\theta_t^{(i)}) + \langle \theta_t^{(i+1)} - \theta_t^{(i)}, \nabla \mathcal{L}(\theta_t^{(i)}) \rangle + \frac{L}{2} \|\theta_t^{(i+1)} - \theta_t^{(i)}\|^2$$
$$= \mathcal{L}(\theta_t^{(i)}) - \eta_t \langle \nabla \hat{\mathcal{L}}(\theta_t^{(i)}), \nabla \mathcal{L}(\theta_t^{(i)}) \rangle + \frac{L\eta_t^2}{2} \|\nabla \hat{\mathcal{L}}(\theta_t^{(i)})\|^2 \tag{20}$$

By the premise of the memory coherence in the theorem and Assumption 1, we have that the inner product of the gradient with and without pending events is bounded by $\mu$, and that the negative sampling is unbiased and has upper bounded variance. Therefore, taking the expectation of each term and substituting the assumption into the expression, we arrive the following inequality,

$$\mathbb{E}[\mathcal{L}(\theta_t^{(i+1)})] \leq \mathbb{E}[\mathcal{L}(\theta_t^{(i)}) - \eta_t \langle \nabla \hat{\mathcal{L}}(\theta_t^{(i)}), \nabla \mathcal{L}(\theta_t^{(i)}) \rangle + \frac{L\eta_t^2}{2} \|\nabla \hat{\mathcal{L}}(\theta_t^{(i)})\|^2]$$
$$\leq \mathcal{L}(\theta_t^{(i)}) - \eta_t \mu \mathbb{E}[\|\nabla \mathcal{L}(\theta_t^{(i)})\|^2] + \frac{L\eta_t^2}{2}(\mathbb{E}[\|\nabla \mathcal{L}(\theta_t^{(i)})\|^2] + \sigma^2) \tag{21}$$
$$= \mathcal{L}(\theta_t^{(i)}) + (-\eta_t \mu + \frac{L\eta_t^2}{2})\mathbb{E}[\|\nabla \mathcal{L}(\theta_t^{(i)})\|^2] + \frac{L\eta_t^2}{2}\sigma^2$$

Rearrange the equation above, we get that

$$(\eta_t \mu - \frac{L\eta_t^2}{2})\mathbb{E}[\|\nabla \mathcal{L}(\theta_t^{(i)})\|^2] \leq \mathcal{L}(\theta_t^{(i)}) - \mathcal{L}(\theta_t^{(i+1)}) + \frac{L\eta_t^2}{2}\sigma^2 \tag{22}$$

Telescope sum over the entire epoch, namely all the batches, $i = 1, ..., K$, we have,

$$\sum_{i=1}^{K} (\eta_t \mu - \frac{L\eta_t^2}{2})\mathbb{E}[\|\nabla \mathcal{L}(\theta_t^{(i-1)})\|^2] \leq \mathcal{L}(\theta_t^{(0)}) - \mathcal{L}(\theta_t^{(K)}) + \sum_{i=1}^{K} \frac{L\eta_t^2}{2}\sigma^2, \tag{23}$$

By definition, we have that $\mathcal{L}(\theta_t^{(K)}) = \mathcal{L}(\theta_{t+1}^{(0)})$. The above equation is equivalent to the following,

$$\sum_{i=1}^{K}(\eta_t\mu - \frac{L\eta_t^2}{2})\mathbb{E}[\|\nabla\mathcal{L}(\theta_t^{(i-1)})\|^2] \le \mathcal{L}(\theta_t^{(0)}) - \mathcal{L}(\theta_{t+1}) + \sum_{i=1}^{K}\frac{L\eta_t^2}{2}\sigma^2, \tag{24}$$

Since we are making the learning rate independent of batch number $i$, we can rearrange the equation above as follows,

$$(\eta_t\mu - \frac{L\eta_t^2}{2})\sum_{i=1}^{K}\mathbb{E}[\|\nabla\mathcal{L}(\theta_t^{(i-1)})\|^2] \le \mathcal{L}(\theta_t^{(0)}) - \mathcal{L}(\theta_{t+1}) + K\frac{L\eta_t^2}{2}\sigma^2, \tag{25}$$

Now telescope sum over the epochs, namely, $t = 1, ..., T$, we have,

$$\sum_{t=1}^{T}(\eta_t\mu - \frac{L\eta_t^2}{2})\sum_{i=1}^{K}\mathbb{E}[\|\nabla\mathcal{L}(\theta_t^{(i-1)})\|^2] \le \mathcal{L}(\theta_0^{(0)}) - \mathcal{L}(\theta_T) + \sum_{t=1}^{T}K\frac{L\eta_t^2}{2}\sigma^2, \tag{26}$$

By set up, we have that $\sum_{i=1}^{K}\mathbb{E}[\|\nabla\mathcal{L}(\theta_t^{(i-1)})\|^2] = \nabla\mathcal{L}(\theta_t^{(0)})$. Then, we can obtain an equivalent expression for the above equation as follows.

$$\sum_{t=1}^{T}(\eta_t\mu - \frac{L\eta_t^2}{2})\mathbb{E}[\|\nabla\mathcal{L}(\theta_t^{(0)})\|^2] \le \mathcal{L}(\theta_0^{(0)}) - \mathcal{L}(\theta_T) + K\sigma^2\frac{L}{2}\sum_{t=1}^{T}\eta_t^2, \tag{27}$$

Note that the choice of stepsize guarantees that

$$\eta_t\mu - \frac{Ls\eta_t^2}{2} > 0$$

for all $t$. Thus, we can divide both side by $\sum_{t=1}^{T}(\eta_t\mu - \frac{L\eta_t^2}{2})$. Rearrange the terms and taking the minimum among $t = 1, ..., T$ for $\|\nabla\mathcal{L}(\theta_t^{(0)})\|^2$, we obtain the following expression,

$$\min_{1\le t\le T}\mathbb{E}[\|\nabla\mathcal{L}(\theta_t^{(0)})\|^2] \le \frac{\mathcal{L}(\theta_0^{(0)}) - \mathcal{L}(\theta_T)}{\sum_{t=1}^{T}(\eta_t\mu - \frac{L\eta_t^2}{2})} + \frac{K\sigma^2\frac{L}{2}\sum_{t=1}^{T}\eta_t^2}{\sum_{t=1}^{T}(\eta_t\mu - \frac{L\eta_t^2}{2})}, \tag{28}$$

By the choice of $\eta_t$, we have that $\eta_t\mu - \frac{L\eta_t^2}{2} > \frac{\eta_t\mu}{2}$, and this leads to,

$$\begin{aligned}\min_{1\le t\le T}\mathbb{E}[\|\nabla\mathcal{L}(\theta_t^{(0)})\|^2] &\le \frac{(\mathcal{L}(\theta_0^{(0)}) - \mathcal{L}(\theta_T))}{\sum_{t=1}^{T}\eta_t\mu/2} + \frac{K\sigma^2\frac{L}{2}\sum_{t=1}^{T}\eta_t^2}{\sum_{t=1}^{T}\eta_t\mu/2}, \\ &\le \frac{2(\mathcal{L}(\theta_0^{(0)}) - \mathcal{L}(\theta_T))}{\sum_{t=1}^{T}\eta_t\mu} + \frac{2K\sigma^2 L\sum_{t=1}^{T}\eta_t^2}{\sum_{t=1}^{T}\eta_t\mu}, \end{aligned} \tag{29}$$

Since $\theta^*$ is the optimal parameter, by definition we have that,

$$\mathcal{L}(\theta^*) \le \mathcal{L}(\theta_T).$$

Substitute this in the equation, we have,

$$\begin{aligned}\min_{1\le t\le T}\mathbb{E}[\|\nabla\mathcal{L}(\theta_t^{(0)})\|^2] &\le \frac{2(\mathcal{L}(\theta_0^{(0)}) - \mathcal{L}(\theta_T))}{\sum_{t=1}^{T}\eta_t\mu} + \frac{2K\sigma^2 L\sum_{t=1}^{T}\eta_t^2}{\sum_{t=1}^{T}\eta_t\mu}, \\ &\le \frac{2(\mathcal{L}(\theta_0^{(0)}) - \mathcal{L}(\theta^*))}{\sum_{t=1}^{T}\eta_t\mu} + \frac{2K\sigma^2 L\sum_{t=1}^{T}\eta_t^2}{\sum_{t=1}^{T}\eta_t\mu}, \end{aligned} \tag{30}$$

Finally, substitute in the stepsize $\eta_t = \frac{\mu}{L\sqrt{Kt}}$, we get that

$$\min_{0\le t\le T}\mathbb{E}[\|\nabla\mathcal{L}(\theta_t)\|^2] \le \left(\frac{2\sqrt{K}L(\mathcal{L}(\theta_0) - \mathcal{L}(\theta^*))}{\mu^2} + \sqrt{K}\sigma^2\log T\right)\frac{1}{\sqrt{T}} \tag{31}$$

$\square$

# D  PROOF OF PRES THEORETICAL GUARANTEE

In this appendix, we provide proofs for the theoretical guarantees of PRES, namely, Proposition 1.

Next, we present the formal version of Propostion 1. It should be noted that the notations used here are slightly different from the notations used in the main paper. We denote that $s_i(t)$ to be the memory state of vertex $i$ at time $t$ when processing events sequentially (i.e., there is no intra-batch dependency). We denote $\hat{s}_i(t)$ to be the memory state of vertex $i$ at time $t$ when pending events are processed in parallel. We denote $s_i(t^-)$ to be the memory state of vertex $i$ from the last update, and similarly for $\hat{s}_i(t-)$.

We assume the difference between these two is given by a zero-mean Gaussian noise, i.e.,

$$s_i(t) = \hat{s}_i(t) + \mathcal{N}(0, \sigma_1). \tag{32}$$

where $\mathcal{N}(0, \sigma_1)$ is the zero-mean Gaussian noise with variance $\sigma_1$. In other words, we have that

$$\delta_{s_i} = s_i(t) - \hat{s}_i(t) = \mathcal{N}(0, \sigma_1).$$

Furthermore, we assume that the state transition for memory state without intra-batch dependency loss is given by the following linear state-space model with Gaussian noise,

$$s_i(t) = s_i(t^-) + (t - t^-)\mathcal{N}(\mu, \sigma_2) \tag{33}$$

Recall that the prediction model we used is,

$$\dot{s}_i(t) = s_i(t^-) + (t - t^-)\hat{\delta}_{s_i} \tag{34}$$

and the final state estimation is given by,

$$\bar{s}_i(t) = \gamma \hat{s}_i(t) + (1 - \gamma)\dot{s}_i(t) \tag{35}$$

**Proposition 2.** *Under the set-up above, we have that*

$$\mathbb{E}[\|\bar{s}_i(t) - s_i(t)\|] \le \mathbb{E}[\|\hat{s}_i(t) - s_i(t)\|].$$

Proposition 2 states that the difference between the memory state estimated with our proposed model and the memory state without intra-batch dependency is smaller than the one with the standard batch processing. This means that our proposed method can mitigate the effect brought by batch processing, as it would be closer to the "true" memory state and therefore has a smaller variance as claimed in Proposition 1. Now, we show that Proposition 2 is true.

*Proof.* First, substitute in

$$\bar{s}_i(t) = \gamma \hat{s}_i(t) + (1 - \gamma)(\dot{s}_i(t)),$$

in

$$\|\bar{s}_i(t) - s_i(t)\|,$$

we have that

$$\begin{aligned} \|\bar{s}_i(t) - s_i(t)\| &= \|\gamma \hat{s}_i(t) + (1 - \gamma)(\dot{s}_i(t)) - s_i(t)\| \\ &= \|\gamma \hat{s}_i(t) - s_i(t) + (1 - \gamma)(\dot{s}_i(t))\| \end{aligned} \tag{36}$$

Notice that if $\gamma = 1$, then we have

$$\begin{aligned} \|\bar{s}_i(t) - s_i(t)\| &= \|\gamma \hat{s}_i(t) - s_i(t) + (1 - \gamma)(\dot{s}_i(t))\| \\ &= \|\hat{s}_i(t) - s_i(t)\| \end{aligned} \tag{37}$$

As $\gamma$ is a learnable parameter between $[0, 1]$, this guarantees that $\|\bar{s}_i(t) - s_i(t)\|$ no worse than $\|\hat{s}_i(t) - s_i(t)\|$. Next, we show that there is indeed a possible improvement. To do so, we substitute

$$s_i(t) = \gamma s_i(t) + (1 - \gamma)s_i(t)$$

into the equation above, and we get

$$\|\bar{s}_i(t) - s_i(t)\| = \|\gamma \hat{s}_i(t) - [\gamma s_i(t) + (1 - \gamma)s_i(t)] + (1 - \gamma)(\dot{s}_i(t))\| \tag{38}$$

Rearranging the terms, we get,

$$\|\bar{s}_i(t) - s_i(t)\| = \|\gamma(\hat{s}_i(t) - s_i(t^-)) + (1 - \gamma)(\dot{s}_i(t) - s_i(t))\| \tag{39}$$

As $\gamma$ is a learnable parameter between $[0, 1]$, as long as we show that

$$(\dot{s}_i(t) - s_i(t)) \mapsto 0,$$

then we have,

$$\begin{aligned}
\|\bar{s}_i(t) - s_i(t)\| &= \|\gamma(\hat{s}_i(t) - s_i(t^-)) + (1 - \gamma)(\dot{s}_i(t) - s_i(t))\| \\
&= \|\gamma(\hat{s}_i(t) - s_i(t^-))\| \\
&\leq \|\hat{s}_i(t) - s_i(t^-)\|
\end{aligned} \tag{40}$$

To show $(\dot{s}_i(t) - s_i(t)) \mapsto 0$, we substitute the modelling assumption and we have,

$$\begin{aligned}
\dot{s}_i(t) - s_i(t) &= s_i(t^-) + (t - t^-)\mathcal{N}(\mu, \sigma_2) - s_i(t^-) + (t - t^-)\hat{\delta}_{s_i} \\
&= (t - t^-)[\mathcal{N}(\mu, \sigma_2) - \hat{\delta}_{s_i}]
\end{aligned} \tag{41}$$

As the number of samples increase, we have that $\mathbb{E}[\hat{\delta}_{s_i}] \mapsto \mathcal{N}(\mu, \sigma_2)$ and therefore,

$$\mathbb{E}(\dot{s}_i(t) - s_i(t)) \mapsto 0.$$

Therefore, we arrive that,

$$\mathbb{E}[\|\bar{s}_i(t) - s_i(t)\|] \leq \mathbb{E}[\|\hat{s}_i(t) - s_i(t)\|].$$

$\square$

# E  EXPERIMENT DETAILS

## E.1  HARDWARE AND SOFTWARE

All the experiments of this paper are conducted on the following machine

CPU: two Intel Xeon Gold 6230 2.1G, 20C/40T, 10.4GT/s, 27.5M Cache, Turbo, HT (125W) DDR4-2933

GPU: four NVIDIA Tesla V100 SXM2 32G GPU Accelerator for NV Link

Memory: 256GB (8 x 32GB) RDIMM, 3200MT/s, Dual Rank

OS: Ubuntu 18.04LTS

## E.2  DATASET

### E.2.1  DESCRIPTION

We use the following public datasets provided by the authors of JODIE Kumar et al. (2019). (1) Wikipedia dataset contains edits of Wikipedia pages by users. (2) Reddit dataset consists of users' posts on subreddits. In these two datasets, edges are with 172-d feature vectors, and user nodes are with dynamic labels indicating if they get banned after some events. (3) MOOC dataset consists of actions done by students on online courses, and nodes with dynamic labels indicating if students drop out of courses. (4) LastFM dataset consists of events that users listen to songs. MOOC and LastFM datasets are non-attributed. The statistics of the datasets are summarized in Table 3.

### E.2.2  LISCENCE

All the datasets used in this paper are from publicly available sources (public paper) without a license attached by the authors.

## E.3  HYPER-PARAMETERS

For the hyper-parameters of each experiment, we follow closely the ones used in TGL Zhou et al. (2022). The only parameter, we control in the experiment is the temporal batch size used for training.

Table 3: Detailed statistic of the datasets.

| Datasets | Wikipedia | Reddit | MOOC | LastFM | GDELT |
|---|---|---|---|---|---|
| # vertices | 9,227 | 10,984 | 7,144 | 1,980 | 16,682 |
| # edges | 157,474 | 672,447 | 411,749 | 1,293,103 | 1,912,909 |
| # edge features | 172 | 172 | 0 | 0 | 186 |

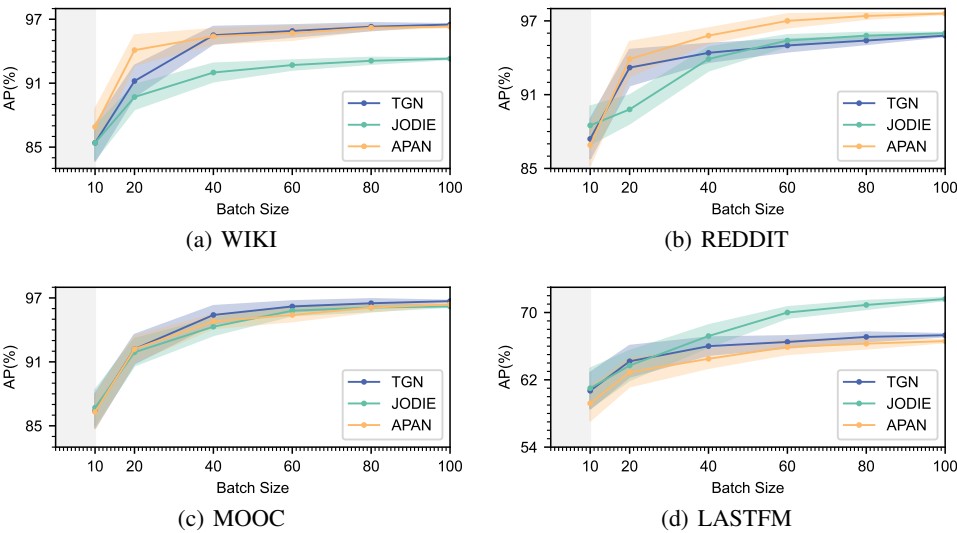

Figure 9: Performance of baseline methods with and without PRES under different batch sizes. The x-axis represents the batch size, while the y-axis represents the average precision (AP). The results are averaged over five trials.

## F    ADDITIONAL RESULTS

In this section, we present additional experimental results.

### F.1    BATCHSIZE AND PERFORMANCE

In this subsection, we provide the additional results on the remaining dataset for the relation between batch size and the performance of MDGNNs. The results are reported in Fig. 9. In this experiment, we fix the default hype parameter provided by TGL and change the temporal batch size used in each dataset.

### F.2    EFFECTIVENESS OF PRES

In this subsection, we provide the additional results on the remaining dataset for the effectiveness of PRES. The results are reported in Fig. 10, Fig. 11, Fig. 12 and Fig. 13. Similar to the last experiment, we fix the default hype parameter provided by TGL and change the temporal batch size used in each dataset, and compare the performance when training with and without PRES.

### F.3    TRAINING EFFICIENCY IMPROVEMENT

In this subsection, we provide the additional results on the remaining dataset for the effectiveness of PRES. The results are reported in Fig. 14. Similar to the last experiment, we fix the default hype parameter provided by TGL as well as the batch size and compare the statistical efficiency when training with and without PRES.

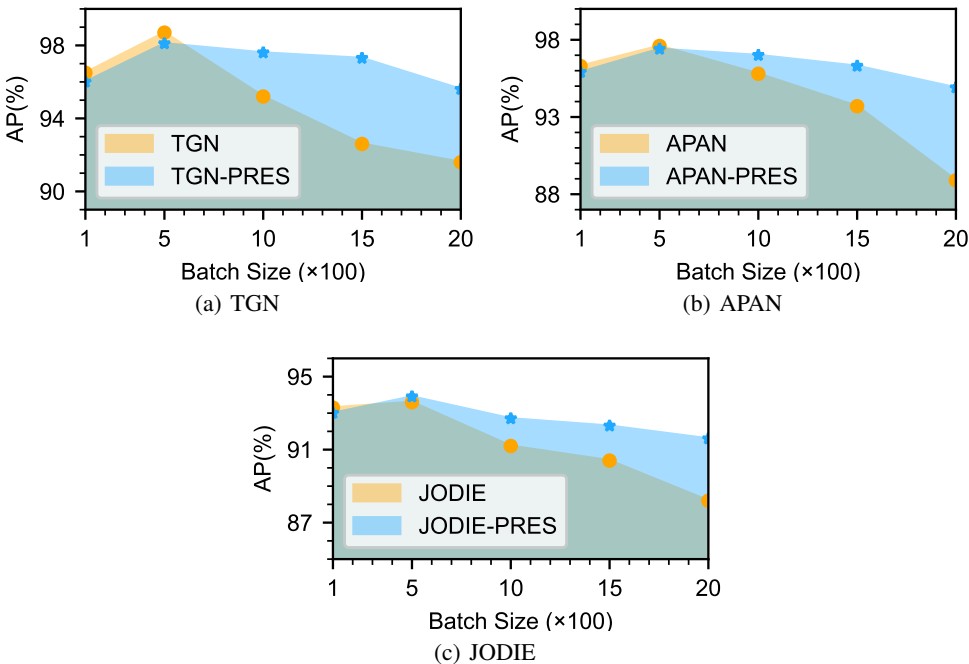

Figure 10: Performance of baseline methods with and without PRES under different batch sizes on WIKI dataset. The x-axis represents the batch size (multiplied by 100), while the y-axis represents the average precision (AP). The results are averaged over five trials with $\beta = 0.1$ for PRES.

### F.4 ADDITIONAL EXPERIMENT TO SHOW COMPARISON WITH OTHER EFFICIENT METHODS FROM OTHER DOMAINS

In this section, we provide additional experimental results to illustrate the relation between relative speedup and the affected performance from the other domains as a comparison.

The studies of efficient method (even at the cost of suboptimal settings for accuracy) is exemplified (but not limited) to the following major lines of research:

1. use of staleness: which sacrifices the "freshness of information" in the training process to accelerate computation or communication in training

2. use of quantization in both training and inference: which sacrifice the precision of the weight of the model to accelerate computation

3. simpler model architecture: use simple estimation method to accelerate the efficiency of the model

We have sampled the following methods from each category to create a comparison between the gain in efficiency and the effect on accuracy: 1) staleness: PipeGCN Wan et al. (2022), SAPipe Chen et al. (2022), Sancus Peng et al. (2022), 2) quantization: AdaQP Wan et al. (2023), 3)simpler architecture: FastGCN Chen et al. (2018).

The values for each method are obtained in the following ways:

- For SAPipe and FastGCN, the values are estimated or taken from the papers. The value of SAPipe is taken from Chen et al. (2022). The underlying tasks for these values are

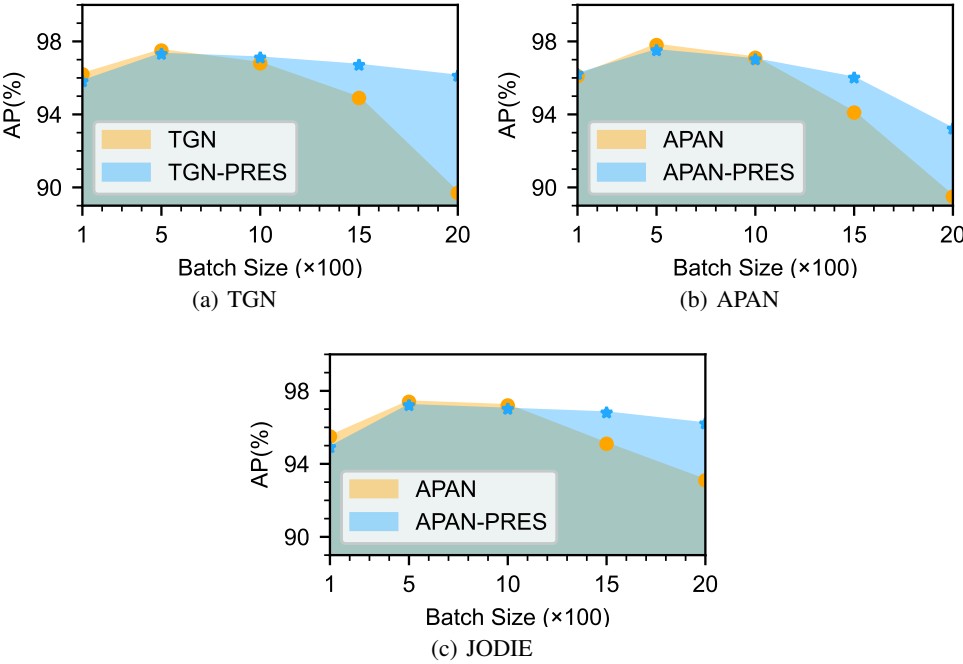

Figure 11: Performance of baseline methods with and without PRES under different batch sizes on REDDIT dataset. The x-axis represents the batch size (multiplied by 100), while the y-axis represents the average precision (AP). The results are averaged over five trials with $\beta = 0.1$ for PRES.

image classification tasks and language translation tasks. The value of FastGCN is taken from Chen et al. (2018) and the underlying task is node classification.

- For Sancus, AdaQP, and PipeGCN, the values are obtained from running the open-source code:
    1. Sancus[1]: node classification with GCN the public OGB-product dataset
    2. AdaQP[2]: node classification with GCN on the public OGB-product dataset
    3. PipeGCN[3]: node classification with GCN on the public OGB-product dataset
- PRES(our) are the average values computed from Table 1.

## F.5 ADDITIONAL EXPERIMENT ON EXTENDED TRAINING SESSION WITH TGN ON WIKI DATASET

## F.6 ADDITIONAL EXPERIMENT ON ABLATION STUDY AND GPU MEMORY UITLIZATION

# G RELATED WORK

In this appendix, we provide a more comprehensive review and discussion of related works.

---

[1]https://github.com/chenzhao/light-dist-gnn

[2]https://github.com/raywan-110/AdaQP

[3]https://github.com/GATECH-EIC/PipeGCN/tree/main

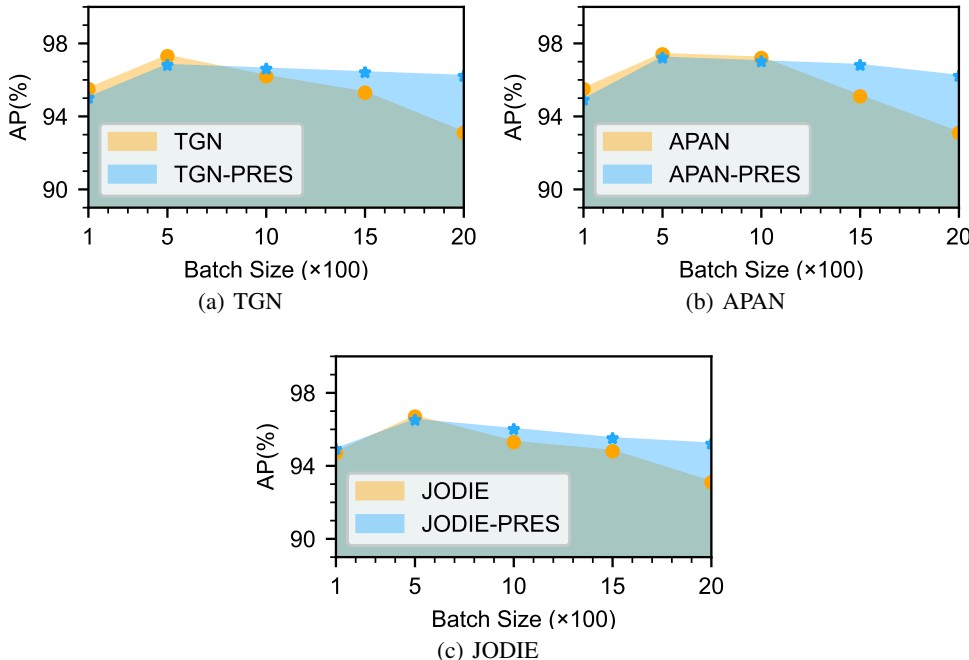

Figure 12: Performance of baseline methods with and without PRES under different batch sizes on MOOC dataset. The x-axis represents the batch size (multiplied by 100), while the y-axis represents the average precision (AP). The results are averaged over five trials with $\beta = 0.1$ for PRES.

### G.1 DYNAMIC GRAPH REPRESENTATION LEARNING

Dynamic graph representation learning has received significant attention in recent years, driven by the need to model and analyze evolving relationships and temporal dependencies within dynamic graphs. Comprehensive surveys Skarding et al. (2021); Kazemi et al. (2020) offer detailed insights into the existing works. Dynamic Graph Neural Networks (DGNNs), as the dynamic counterparts of GNNs, have emerged as promising neural models for dynamic graph representation learning (Sankar et al., 2020; Poursafaei et al., 2022; Xu et al., 2020; Rossi et al., 2021; Wang et al., 2021; Kumar et al., 2019; Su et al., 2024; Trivedi et al., 2019; Zhang et al., 2023; Pareja et al., 2020; Trivedi et al., 2017). Among DGNNs, MDGNNs such as (Rossi et al., 2021; Wang et al., 2021; Kumar et al., 2019; Trivedi et al., 2019; Zhang et al., 2023) have demonstrated superior inference performance compared to their memory-less counterparts. Most existing works on MDGNNs primarily focus on designing tailored architectures for specific problems or settings. To the best of our knowledge, only a few studies have investigated the efficiency of MDGNNs. For example, Wang & Mendis (2023) leverages computation redundancies to accelerate the inference performance of the temporal attention mechanism and the time encoder. Wang et al. (2021) introduces a mailbox that stores neighbour states for each node, accelerating local aggregation operations during online inference. Zhou et al. (2022); Sheng et al. (2024) provides a system framework for optimizing inter-batch dependency and GPU-CPU communication during MDGNN training. Zhang et al. (2023) proposes a restarter module that focuses on restarting the model at any time with warm re-initialized memory. EDGE (Chen et al., 2021) focuses on accelerating event embedding computation by intentionally neglecting some updates (i.e., using staleness). Therefore, EDGE's focus and objectives differ from ours. DistTGL (Zhou et al., 2023) focuses on the dependency arising from distributed training and addresses efficient scaling concerning the number of GPUs. While these works have addressed the

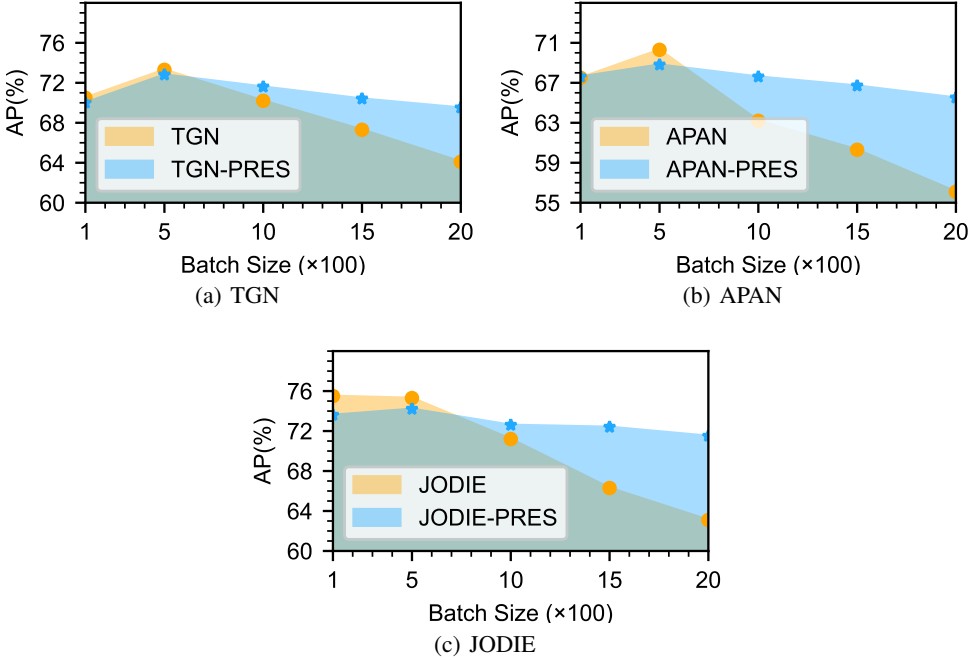

Figure 13: Performance of baseline methods with and without PRES under different batch sizes on LASTFM dataset. The x-axis represents the batch size (multiplied by 100), while the y-axis represents the average precision (AP). The results are averaged over five trials with $\beta = 0.1$ for PRES.

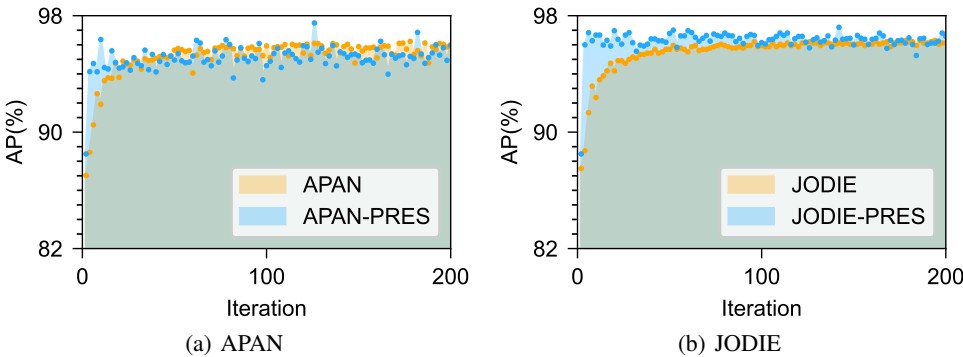

Figure 14: Statistical efficiency of baseline method w./w.o PERS. x-axis is the training iteration and y-axis is the average precision. $\beta = 0.1$ is used in PRES.

efficiency of MDGNNs in some aspects, none of them provide theoretical insights or directly address the intra-batch dependency problem. In contrast, our study focuses on enlarging the temporal batch size to enable better data parallelism in MDGNN training. We adopt a more theoretical approach, and our proposed framework can be used in conjunction with these previous works to further enhance training efficiency. It should be noted that there exists another line of research concerning updating the GNN models on dynamic/expanding graphs, referred to as graph continual learning (Su et al., 2023). The objective of this line of research is orthogonal to dynamic representation learning.

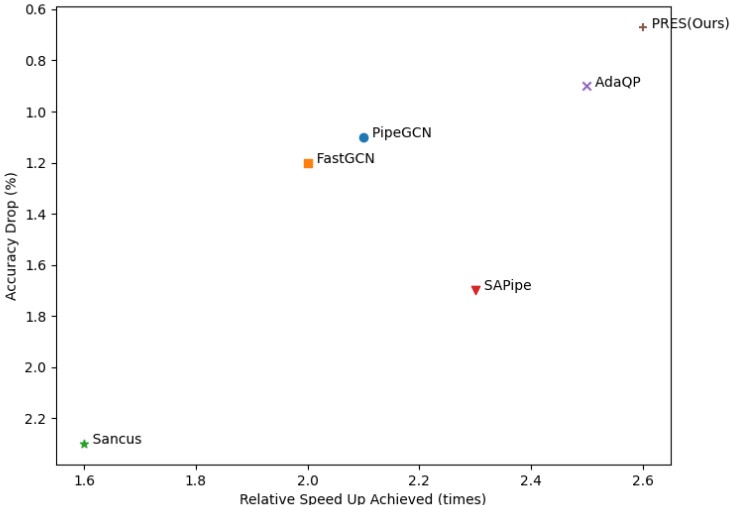

Figure 15: The visualization of the relative speed and affected accuracy of methods from different domains. Note that the value of the y-axis is reversed. The points with higher positions in the y-axis have a smaller accuracy drop. In addition, these methods are from different domains, corresponding to different underlying tasks and datasets. This figure is only a rough comparison, as rigorously they are not directly comparable because of different tasks and datasets. However, this figure demonstrates that the accuracy-speed trade-off of PRES(our) is reasonable

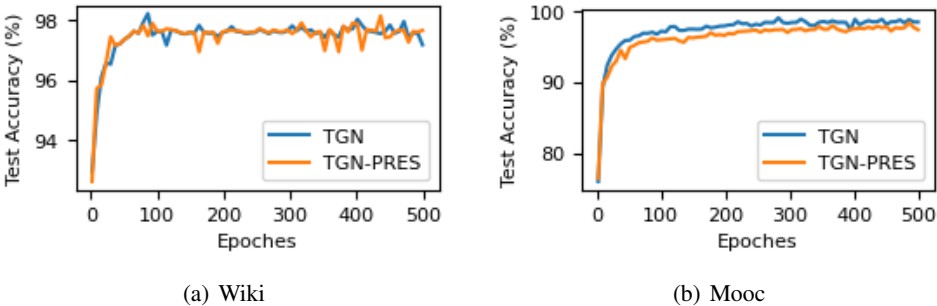

(a) Wiki            (b) Mooc

Figure 16: Extended training sessions with TGN on WIKI and MOOC Datasets. Figure 16(a) presents the results of the TGN model with 500 epochs on the WIKI dataset, and Figure 16(b) presents the results of the TGN model with 500 epochs on the MOOC dataset. Fig. 16 (a) illustrates that with significantly extended training sessions, many of the minor accuracy discrepancies observed in datasets like Wiki between TGN and TGN-PRES can be alleviated or attributed to fluctuations arising from distinct fitting processes. Additionally, Fig. 16 (b) demonstrates that the discrepancy gap on certain datasets, such as MOOC, gradually diminishes as training progresses.

## G.2 MINI-BATCH IN STOCHASTIC GRADIENT DESCENT (SGD)

Another line of research investigates the effect of mini-batch size in SGD training (Goyal et al., 2017; Qian & Klabjan, 2020; Lin et al., 2018; Akiba et al., 2017; Gower et al., 2019; Woodworth et al., 2020; Bottou et al., 2018; Schmidt et al., 2017). Research on the relation between mini-batch size and SGD has been an active area of investigation in the field of deep learning. The choice of mini-batch size plays a crucial role in balancing the computational efficiency, convergence speed, and generalization performance of neural networks. Larger mini-batches tend to provide more accurate gradient estimates due to increased sample size, resulting in faster convergence. However,

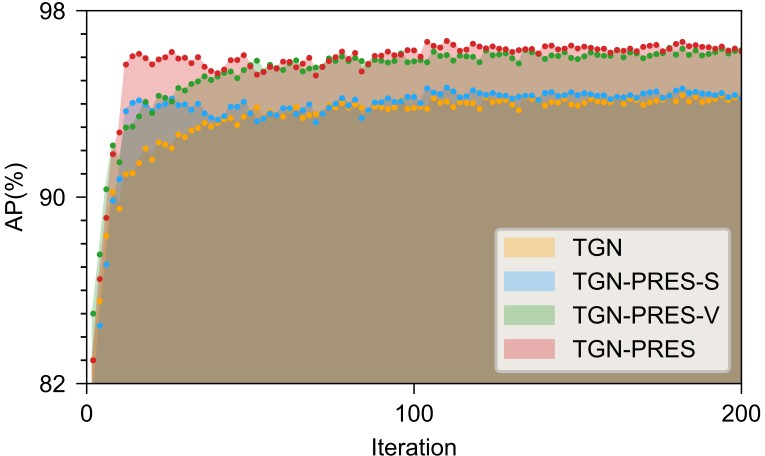

Figure 17: Ablation study of the PRES framework. TGN represents the standard implementation devoid of any PRES component. TGN-PRES-S is TGN augmented with memory coherence smoothing only. TGN-PRES-V is TGN implemented solely with the prediction-correction scheme. TGN-PRES is TGN combined with both memory coherence smoothing and the prediction-correction scheme. The results validate the distinct operational nature of the two proposed techniques. Memory coherence smoothing enhances the convergence rate of the learning process, while the prediction-correction scheme attenuates the variance introduced by a large temporal batch size. The experiment was conducted on the WIKI dataset with a batch size of $1000$ and a $\beta = 0.1$ for the PRES scheme.

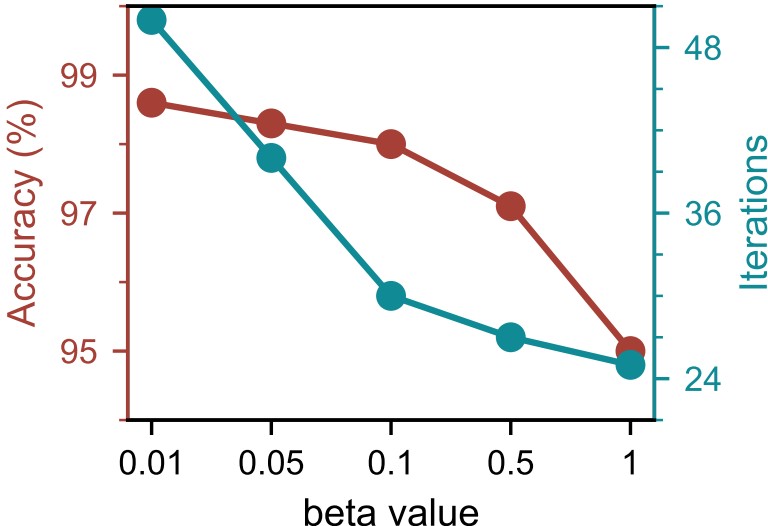

Figure 18: Ablation study of the $\beta$ value in PRES framework. Fig. 18 illustrates that increasing $\beta$ leads to faster convergence but can have a negative effect on the accuracy. Because of this trade-off, $\beta$ can not be too large nor too small. The plot above motivates our choice of $\beta = 0.1$ in the experiments.

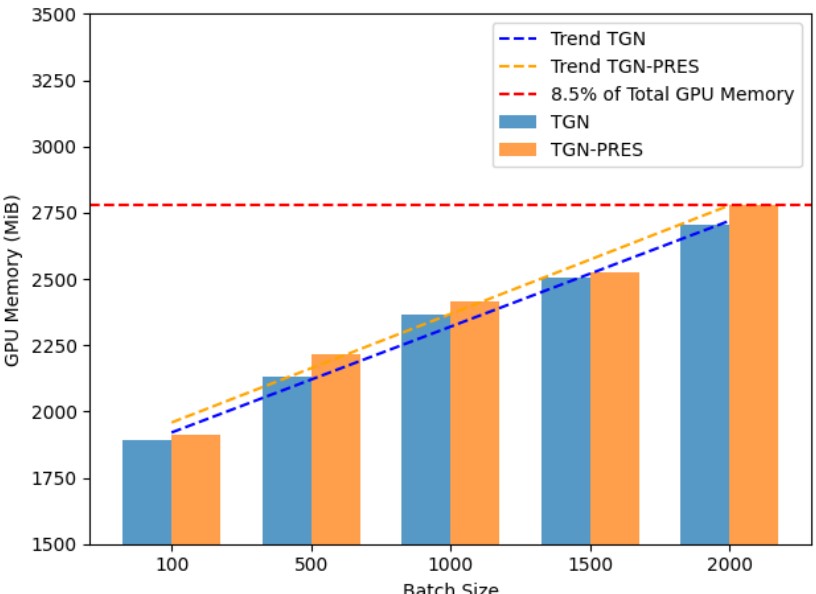

Figure 19: This illustration presents the GPU memory utilization of baseline methods, both with and without the PRES implementation, under varying batch sizes on the WIKI dataset. The x-axis denotes the batch size, while the y-axis depicts the GPU memory usage. The findings reveal that the additional GPU memory required by PRES is not influenced by the batch size, indicating its scalability with increasing batch sizes. Moreover, the red line demonstrates significant underutilization of the GPU memory in training MDGNNs.

they also require more memory and computational resources. On the other hand, smaller mini-batches can introduce more noise into the gradient estimates but may provide better generalization by exploring a more diverse set of examples. Researchers have explored the impact of mini-batch size on convergence properties, optimization dynamics, and generalization performance, leading to various recommendations and insights. However, it is important to differentiate the concepts of mini-batch in SGD and temporal batch in MDGNNs, as they serve distinct purposes and bring different challenges. The goal of mini-batches in SGD is to obtain a good estimation of the full-batch gradient by downsampling the entire dataset into mini-batches. On the other hand, the temporal batch specifically refers to partitioning consecutive graph data to ensure the chronological processing of events. The temporal batch problem studied in this paper aims to increase the temporal batch size to enhance data parallelism in MDGNN training.

### G.3 SAMPLING IN GNNS

The full-batch training of a typical GCN is employed in Kipf & Welling (2017) which necessities keeping the whole graph data and intermediate nodes' representations in the memory. This is the key bottleneck that hinders the scalability of full-batch GCN training. To overcome this issues, research on vertices and neighbor sampling in graph neural networks (GNNs) has been a topic of significant interest in the field of graph representation learning (Chen et al., 2017; Ying et al., 2018; Huang et al., 2018; Goyal et al., 2017; Qian & Klabjan, 2020; Lin et al., 2018; Akiba et al., 2017; Gandhi & Iyer, 2021; Hamilton et al., 2017; Chen et al., 2018; Zou et al., 2019). GNNs operate on graph-structured data and aim to capture the relational information among nodes. One crucial aspect is selecting which nodes to consider during the learning process. Vertices sampling refers to the selection of a subset of nodes from the graph for computational efficiency, as processing the entire graph can be computationally expensive for large-scale networks. Different sampling strategies have been explored, including random sampling, stratified sampling based on node properties or degrees, and importance sampling based on node importance measures. On the other hand, neighbour sampling focuses on determining which neighbours of a node to consider during the aggregation step in GNNs.

It addresses the challenge of scalability by only considering a subset of a node's neighbours in each layer, reducing the computational complexity. Various neighbour sampling techniques have been proposed, such as uniform sampling, adaptive sampling based on node degrees, or sampling proportional to node attention scores. Researchers have investigated the effects of different sampling strategies on the expressiveness, efficiency, and generalization capabilities of GNNs, aiming to strike a balance between computational efficiency and capturing important graph structures. Essentially, the problem studied in GNN sampling is similar to the study of mini-batch in SGD, which is to efficiently and effectively estimate the gradient with sampling. Therefore, it is also orthogonal to the problem we studied here.

