# OpenReview forum: "PRES: Toward Scalable Memory-Based Dynamic Graph Neural Networks"
_ICLR.cc/2024/Conference — ICLR 2024 poster_

### Official Review · Reviewer_pqbr · 2023-10-31

**Soundness:** 3 good
**Presentation:** 3 good
**Contribution:** 3 good
**Rating:** 6
**Confidence:** 3

**Summary:**

This paper aims to offer a scalable training method for Memory-Based Dynamic Graph Neural Networks (MDGNNs) by mitigating the temporal discontinuity issue, thus training MDGNNs with large temporal batch sizes. It consists of two main contributions: 1) conducting a theoretical study on the impact of temporal batch size on the convergence of MDGNN training, and 2) proposing PRES based on the theoretical study, an iterative prediction-correction scheme combined with a memory coherence learning objective to mitigate the effect of temporal discontinuity. The evaluation shows that the proposed approach enables up to 4X larger temporal batch sizes and achieves up to 3.4X speedup during MDGNN training.

**Strengths:**

+ It targets an emerging and important GNN, MDGNNs, and the proposed designs generally make sense.
+ The problem definition is easy-to-follow.
+ The introduced concept of memory coherence is interesting.
+ The code is publicly available.

**Weaknesses:**

- The trained graphs seem small. It is unclear how PRES performs on large-scale graphs.
- No absolute execution time is reported.
- It is tested on four GPUs. Its scalability to multi-nodes (with more GPUs) is somewhat unclear.
- In many cases, PRES still sacrifices some precision for performance gains.

**Questions:**

Overall, this is a solid study with clear innovations. The theoretical study on the impact of temporal batch size on the convergence of MDGNN training is extensive and helpful. My major concerns focus on the evaluation aspects. It would be extremely helpful if the authors could offer more information about these questions:

1. PRES is mainly evaluated on four graph datasets (Reddit, Wiki, Mooc, and LastFM). It seems these graphs are not very large with around 1K to 10K vertices and 400K to 1.3M edges. It would be helpful to justify that these graphs are large enough or PRES’s performance is not affected by the graph size.

2. It would be helpful to report the absolute execution time as well rather than relative speedup only.

3. It would be helpful to discuss if this method can be extended to multi-nodes with more GPUs.

4. It seems Table 1 shows that PRES still sacrifices some precision for performance gains in many cases. Please correct me if I have any misunderstanding here.

---

> ### Author Response · Authors · 2023-11-16
> **Response 1**
>
> Thank you for your detailed feedback and comments. Based on your feedback, we have revised the paper and would like to address and clarify your concerns as follows:
>
> **Q1.** PRES is mainly evaluated on four graph datasets (Reddit, Wiki, Mooc, and LastFM). It seems these graphs are not very large with around 1K to 10K vertices and 400K to 1.3M edges. It would be helpful to justify that these graphs are large enough or PRES’s performance is not affected by the graph size.
>
> - MDGNN primarily employs a link prediction task, which makes dataset size mostly dependent on the number of events rather than vertices. The datasets and baselines used in our empirical study adhere to the standard practices within the MDGNN field (e.g., [1][2]).
> - The analysis and methods presented in the paper primarily focus on the size of the temporal batch. They are independent of the size of the dataset. As a result, they are applicable to datasets of any scale.
> - We have included an additional dataset, GDELT, containing approximately 17k vertices and 2 million events in the main experiment (page 8). The results from this dataset align with our predictions and expectations.
>
> | GDELT | TGN | JODIE | APAN |
> | --- | --- | --- | --- |
> | without PRES | 96.8%, 50 Epoches, 1325 Second/Epoch | 95.1%,  50 Epoches, 1123 Second/Epoch | 96.7%,   50 Epoches, 1215 Second/Epoch |
> | with PRES | 96.0%,   50 Epoches, , 490 Second/Epoch,  | 94.3%,  50 Epoches, 401 Second/Epoch | 96.0%,  50 Epoches, 506 Second/Epoch |
>
> **Q2. It would be helpful to report the absolute execution time as well rather than relative speedup only.**
>
> - Thank you for the suggestions. We have included absolute execution time in the revised version (Table 1 Page 8). The absolute execution time reported aligns with the value of the observed speed-up.
>
> **Q3. It would be helpful to discuss if this method can be extended to multi-nodes with more GPUs.**
>
> - Our method and analysis apply to the temporal batch size of MDGNN, which can be viewed as the global batch size in distributed deep neural network training. Therefore, our method can naturally be applied in a multi-node setting to enable a larger "temporal (global) batch size" (data parallelism) and better leverage the abundant computational resources available in such configurations.
> - It is worth noting that in the current state, MDGNN struggles to fully harness the abundant computational resources offered in a multi-node setup, due to the limited temporal batch size. Addressing this limitation forms a central motivation behind our work, and we aspire to enhance MDGNN's efficiency in capitalizing on computational resources in the future.
>
> **W4.  It seems Table 1 shows that PRES still sacrifices some precision for performance gains in many cases. Please correct me if I have any misunderstanding here.**
>
> - We want to clarify that the slight drop in model accuracy shown in Table 1 is very small, around 1%. This trade-off between making the training process faster and maintaining high accuracy is quite beneficial, especially during the phase when developers are experimenting with different settings for their models (like trying different model types or regularization techniques). When we achieve a speedup of around 2 to 3 times, it significantly speeds up the development process, saving a lot of time and possibly money if we're using cloud services. This speed boost allows developers to try out more configurations and methods, which can help them find the best settings and make up for the tiny accuracy loss.
> - It's important to note that the broader community also values training efficiency, as seen in various studies that explore methods to improve efficiency even if it means a small drop in accuracy (like the line of research on using staleness in training [3]).
>
> Thank you once more for the invaluable insights. We hope that our response has addressed your concerns and provided clarifications that further emphasize the significance and contributions of our work.
>
> [1] Zhang, Yao, et al. "TIGER: Temporal Interaction Graph Embedding with Restarts." *Proceedings of the ACM Web Conference 2023*. 2023.
>
> [2] Emanuele Rossi, Ben Chamberlain, Fabrizio Frasca, Davide Eynard, Federico Monti, and Michael Bronstein. Temporal Graph Networks for Deep Learning on Dynamic Graphs. In Proceedings of International Conference on Learning Representations, 2021
>
> [3] Wan, Cheng, et al. "PipeGCN: Efficient Full-Graph Training of Graph Convolutional Networks with Pipelined Feature Communication." *International Conference on Learning Representations*. 2021.

---

> ### Author Response · Authors · 2023-11-19
>
> Thank you again for the feedback on our paper. We hope that our responses have addressed your inquiries and concerns. If this is not the case, please inform us and we would be glad to engage in further discussion.

---

> > ### Comment · Reviewer_pqbr · 2023-11-22
> >
> > I sincerely appreciate the authors' careful response to my questions. Most of my questions are addressed. However, the absolute execution time somewhat confirmed my concern about the problem size. In addition, after reading another review, it is difficult for me to champion this paper. I want to maintain my score.

---

> > > ### Author Response · Authors · 2023-11-23
> > >
> > > Thank you for the reply. We are pleased to note that our rebuttal has effectively addressed most of your concerns.
> > >
> > > We would like to clarify that the absolute time values in the updated table correspond to the time required for a single epoch. The complete training session, such as the one for the GDELT dataset, takes approximately 18 hours to complete which, we believe, qualifies as a significant time investment for training on a large dataset.
> > >
> > > If you have any additional concerns or questions, please do not hesitate to reach out to us.

---

### Official Review · Reviewer_dnLR · 2023-11-01

**Soundness:** 2 fair
**Presentation:** 3 good
**Contribution:** 3 good
**Rating:** 6
**Confidence:** 3

**Summary:**

This work conducts a theoretical study on the impact of the temporal batch size in MDGNN training. This shows that there can be a significant gradient variance using a small temporal batch, which in turn sheds light on an unexpected benefit of large batch sizes. Next, the authors define memory coherence, which represents the similarity of gradient directions within a temporal batch. Memory coherence is then used to model the upper boundary of gradient.
With these theoretical insights, the authors present PRES with two main components: 1) iterative prediction-correction scheme 2) memory coherence smoothing. The former uses a GMM (updated with MLE) to predict newest memory states and fuses it with the calculated memory state to obtain the final ‘corrected’ state. The latter uses a new learning objective to promote larger memory coherence.
Using PRES, the authors were able to increase the temporal batch size without compromising overall accuracy.

**Strengths:**

-	The authors provide theoretical results on the influence of temporal batch size on MDGNN training.
-	With memory coherence, the authors effectively define new methods to compensate for the accuracy drop of naively increasing the temporal batch size.

**Weaknesses:**

- The tradeoff of improved speed at the cost of lower accuracy does not seem to be appealing.
- Comparison with prior work on increasing temporal batch size is insufficient.
-	In a similar manner, there are only a small number of baselines in the experimental results.
-	The specific results of a dataset (LASTFM) is excluded.

**Questions:**

The paper is overall well written. The introduction on MDGNN was easy to follow. Insights from theoretical analyses were well presented. It was also evident how these insights became the main building blocks of PRES. However, my main concern comes from the experiment section.

-	**Is it really useful to gain speed at the cost of accuracy?**

So far this is my main concern. At first I thought the authors were trying to achieve SOTA accuracy.
However, what the authors are doing is gaining speedup of around 2x to 3x, at the cost of decreased accuracy (~1.0%).
I am not so sure about this, considering the effort the community is putting to gain higher accuracy.
Especially on the tested datasets, the number of vertices is only around few thousands, which wouldn't take terribly long to train.
I believe this partially comes from not reporting the training time (only the speedup is reported) and there are not enough baselines to compare. But the bottomline is that a strong justification is needed for this.

-	**What is the consensus on the ‘optimal’ temporal batch size?**

	In Figure 3, the authors show the performance of baselines by increasing the batch size up to 100. In the figure the ‘small batch size’ seems to be ~50. My question is do the majority of MDGNNs use a batch size smaller than 50, or are they already using approximately 500 (which seems to be the optimal size in Figure 4)? If the latter is the case, then personally the insight from theorem 1 (variance of the gradient for the entire epoch can be detrimental when the temporal batch size is small) loses some of its shine. Thus, the authors should try to first do a comprehensive overview on the currently used batch sizes.

-	**How does PRES differ from other baselines?**
	Two related works came to my mind which are missing in the current paper. “Efficient Dynamic Graph Representation Learning at Scale” (arXiv preprint, 2021, https://arxiv.org/abs/2112.07768) and “DistTGL: Distributed Memory-Based Temporal Graph Neural Network Training” (arXiv preprint, 2023, https://arxiv.org/abs/2307.07649). Both try to increase the temporal batch size without harming the accuracy. The former also uses prediction to utilize data-parallelism, while the latter tries to push the temporal batch size to the extreme for distributed GPU clusters. In my opinion, both (and any other baseline that shares the same goal with this work) should be compared methodologically and speedup-wise (in the current setting). Also, it would be interesting to see if these can also benefit from PRES.

-	**Why is performance with/without PRES not shown with the LASTFM dataset?**
LASTFM stands out in that 1) the AP is the lowest 2) the speedup of PRES is the lowest. However, I was unable to find a figure like figure 4 for this dataset. Is there a reason for only leaving this dataset out?

---

> ### Author Response · Authors · 2023-11-16
> **Response 1**
>
> Thank you for your detailed feedback and comments. Based on the feedback, we have revised the paper and would like to address and clarify the concerns as follows:
>
> **Q1 (W1). Usefulness of improved training efficiency and the scale of dataset (tradeoff of improved speed)**
>
> **Usefulness and Significance of Improve Training Efficiency**
>
> - Enhancing training efficiency is particularly crucial in the development phase, where it is necessary to experiment with various hyperparameter settings (e.g., different model configurations, and different regularizations). Achieving a speedup of approximately 2x to 3x can significantly accelerate the development process, resulting in substantial savings of computation, waiting time, and potentially monetary cost when utilizing cloud services. This acceleration empowers developers to explore a broader range of configurations and methodologies, ultimately leading to the potential discovery of optimal settings that can offset the minor accuracy trade-offs (~1%).
> - The importance of training efficiency is also well-recognized and of interest to the community, as evidenced by numerous studies exploring techniques to increase efficiency despite some loss in accuracy (e.g., the extensive line of research in using staleness in training [1]).
>
> **Scale of the Tested Dataset and Report of Training Time**
>
> - We have reported the training time of a complete epoch for each dataset and model (Table 1 of Page 8) in the revised version.  For instance, when training TGN with default settings (50 epoches) on the LASTFM dataset can take up to 9 hours within our computational environment (NVIDIA Tesla V100). Therefore, achieving the observed speed improvements can result in substantial savings (~5.8 hours) in computation and waiting time.
> - MDGNN employs a link prediction task for training, making the size of the dataset dependent on the number of events rather than vertices. The datasets utilized in our study encompass event ranges from 10k (WIKI) to 1.3 million (LASTFM), which are reasonably large for academic purposes. Additionally, we introduced an additional dataset, GDELT, consisting of approximately 17k vertices and 2 million events in the main experiment (as shown in Table 1 on Page 8 in the revised paper). The results for this dataset align with the observations from the other datasets:
>
> | GDELT | TGN | JODIE | APAN |
> | --- | --- | --- | --- |
> | without PRES | 96.8%, 50 Epoches, 1325 Second/Epoch | 95.1%,  50 Epoches, 1123 Second/Epoch | 96.7%,   50 Epoches, 1215 Second/Epoch |
> | with PRES | 96.0%,   50 Epoches, , 490 Second/Epoch,  | 94.3%,  50 Epoches, 401 Second/Epoch | 96.0%,  50 Epoches, 506 Second/Epoch |
>
> **Q2. What is the consensus on the ‘optimal’ temporal batch size? and significance of Theorem 1**
>
> - First, we would like to point out that the batch size of 500 is still small relative to the size of the dataset. Furthermore, the choice of temporal batch size in practical applications should not diminish the significance of Theorem 1. The main contribution of Theorem 1 is to provide an alternative perspective to the conventional wisdom regarding the use of large batches. The prevailing belief has been that employing a large temporal batch size is generally detrimental due to the interdependence of events within that batch. However, Theorem 1 uncovers a surprising advantage of using a large temporal batch size, which results in a smaller variance in the gradient.
> - The impact of temporal batch size represents a relatively uncharted territory within MDGNN research, which constitutes a central motivation for our work. Due to the trade-off between training efficiency and sensitivity to accuracy, determining the "optimal batch size" is likely to vary for different datasets and tasks. While establishing a unified criteria for determining the optimal temporal batch size is an interesting research topic, it falls beyond the primary scope of this paper.

---

> > ### Author Response · Authors · 2023-11-16
> > **Response 1 (continued)**
> >
> > **Q3 (W2,W3). Comparison with previous baselines and how does PRES differ from [4] [5]?**
> >
> > - As previously mentioned, the impact of temporal batch size represents a relatively uncharted territory within MDGNN research, forming a central motivation for our work. We are not aware of any existing works that have explicitly focused on increasing the temporal batch size in MDGNN training.
> > - The primary goal of our empirical study is to demonstrate that our method can expedite the training of existing MDGNN models. The baseline methods we employ are standard and representative within the field of MDGNN, as evidenced by prior research [2][3].
> >
> > Thank you for pointing out the potential omission of related works. We have now included a discussion of these two papers in the extended related works section in the revised paper. Here is a brief summary of the distinctions:
> >
> > - EDGE [4]: EDGE focuses on accelerating computation by being selective to the event updates while our focus is to enlarge temporal batch size. Therefore, EDGE's focus and objectives differ from ours. Furthermore, we are unable to rapidly implement EDGE for direct comparison as they did not provide the public code.
> > - DistTGL [5]: Our method and problem are orthogonal to the issues addressed by DistTGL. Therefore, our method can naturally be used in conjunction with DistTGL.
> >     - DistTGL endeavours to construct an efficient pipeline for a distributed MDGNN training system. It focuses on developing solutions to handle dependencies that emerge across different batches during distributed training. Additionally, DistTGL addresses the challenge of efficient scaling concerning the number of GPUs.
> >     - In contrast, our analysis is centred on the interdependencies of events within a batch and investigates the implications of temporal batch size. Our method aims to enlarge the size of the temporal batch without suffering significant performance degradation.
> >     - We will include the experiment that combines our method and DistTGL to explore whether the performance can be further improved.
> >
> >
> > **Q4 (W4)**. **The Performance with/without PRES not shown with the LASTFM dataset?**
> >
> > - We appreciate your observation regarding the absence of figures for the LASTFM dataset. In response, we have included the "performance with/without PRES" for the LASTFM dataset in Appendix F of the revised paper.
> > - The figure in the appendix illustrates that the baselines with PRES experience only minimal performance degradation as the batch size increases. In contrast, the baselines without PRES exhibit significant performance deterioration. These findings are consistent with our observations on other datasets.
> >
> > Thank you once more for the invaluable insights. We hope that our response has addressed your concerns and provided clarifications that further emphasize the significance and contributions of our work.
> >
> > [1] Wan, Cheng, et al. "PipeGCN: Efficient Full-Graph Training of Graph Convolutional Networks with Pipelined Feature Communication." *International Conference on Learning Representations*. 2021.
> >
> > [2] Zhang, Yao, et al. "TIGER: Temporal Interaction Graph Embedding with Restarts." *Proceedings of the ACM Web Conference 2023*. 2023.
> >
> > [3] Emanuele Rossi, Ben Chamberlain, Fabrizio Frasca, Davide Eynard, Federico Monti, and Michael Bronstein. Temporal Graph Networks for Deep Learning on Dynamic Graphs. In Proceedings of International Conference on Learning Representations, 2021
> >
> > [4] Chen, Xinshi, et al. "Efficient Dynamic Graph Representation Learning at Scale." *arXiv preprint arXiv:2112.07768* (2021).
> >
> > [5] Zhou, Hongkuan, et al. "DistTGL: Distributed Memory-Based Temporal Graph Neural Network Training." *Proceedings of the International Conference for High Performance Computing, Networking, Storage and Analysis*. 2023

---

> > > ### Comment · Reviewer_dnLR · 2023-11-21
> > > **thanks for the response**
> > >
> > > Thanks for the response.
> > > - I am little bit more, but still not fully convinced on the claim between accuracy vs speedup. This is because many GNN (and TGN) papers propose around 1% improvement in each publication. The authors mention that PipeGCN [1] utilizes staleness, but PipeGCN does not sacrifice accuracy at all.
> > > - Maybe one way to make this work shine is to provide a 2D plot of speed-accuracy comparison among a series of other work to show the provided trade-off is meaningful. That is, something we could find from pruning/quantization/nas papers a few year ago (although those were usually comparing the inference time with accuracy). For example, Fig1 of Efficientnet.
> > >
> > > - I could not find LastFM result from appendixF of the revision. Could you provide a precise pointer?

---

> ### Author Response · Authors · 2023-11-19
>
> Thank you again for the feedback on our paper. We hope that our responses have addressed your inquiries and concerns. If this is not the case, please inform us and we would be glad to engage in further discussion.

---

> ### Author Response · Authors · 2023-11-21
>
> First of all, thank you for the response, suggestions, and the chance to further clarify and address your concerns.
>
> > I am little bit more, but still not fully convinced on the claim between accuracy vs speedup. This is because many GNN (and TGN) papers propose around 1% improvement in each publication. The authors mention that PipeGCN [1] utilizes staleness, but PipeGCN does not sacrifice accuracy at all.
>
> We would like to further address the significance of improved training efficiency with the following aspects:
>
> Motivating setting and scenario
>
> - We agree that accuracy is of central importance to machine learning models under unlimited resources. However, different settings might have different preferences. For instance,  a setting with limited resources (time or computation) might prefer better training efficiency.
>
>     One typical application of MDGNN, the mode considered in our paper,  is an E-commerce platform where MDGNN is used for real-time personalization of user experience [2]. In such a scenario, models must be trained within tight time constraints to adapt to evolving user behaviour and preferences, ensuring that recommendations remain relevant and engaging.
>
>
> PipeGCN
>
> - First of all, we would like to point out that it takes a large epoch number (e.g., training Reddit for 2000 epochs, a significantly higher number compared to the typical 100 to 200 epochs) for PipeGCN to reach the same performance as the baselines. In our experimental setting, we follow the default setup as TGN which uses 50 epochs. As illustrated by Fig.4 [1], PipeGCN also experiences minor performance degradation (~1%) if we look at the region where the vanilla baseline has converged when compared to the vanilla baseline in the convergence region (e.g., around 250 epochs for ogbn-products and 1000 epochs for Reddit).
>
>     As our proposed method does not affect the expressive power of the underlying models,  we believe our method can also achieve similar experimental results as PipeGCN if we allow the training session to be long enough. We are working hard on showing such an experiment and will have you updated once we have the data ready.
>
>
> > Maybe one way to make this work shine is to provide a 2D plot of speed-accuracy comparison among a series of other works to show the provided trade-off is meaningful. That is, something we could find from pruning/quantization/nas papers a few year ago (although those were usually comparing the inference time with accuracy). For example, Fig1 of Efficientnet.'
>
> - Thank you for suggesting a 2D plot to showcase the trade-off between speed and accuracy among various works. We are diligently working on creating such a plot and will notify you as soon as we have both the data and plot ready.
>
> > I could not find LastFM result from Appendix F of the revision. Could you provide a precise pointer?
>
> - The figures of the LastFM dataset are included in Figure 13 Page 24 of the updated version.
>
> We sincerely appreciate your engagement and feedback, which contribute to the improvement of our work.
>
> [1] Wan, Cheng, et al. "Pipegcn: Efficient full-graph training of graph convolutional networks with pipelined feature communication." *arXiv preprint arXiv:2203.10428* (2022).
>
> [2] Bai, Ting, et al. "Temporal graph neural networks for social recommendation." *2020 IEEE International Conference on Big Data (Big Data)*. IEEE, 2020.

---

> > ### Comment · Reviewer_dnLR · 2023-11-21
> > **reply**
> >
> > Thanks for the response. I will be waiting for the results (DistTGL, speed-accuracy comparison, long enough training)
> > - I still cannot see the result on page 24. could you check again if the revision has been correctly uploaded?
> > - On PipeGCN, could you check if we are looking at the same curve? I believe we should be comparing GCN vs PipeGCN-GF and they both have converged at 250 epochs.

---

> ### Author Response · Authors · 2023-11-23
> **Response (1/2)**
>
> Thank you once again for your responsiveness and valuable suggestions. We greatly appreciate your feedback and the opportunity to address your concerns.
>
> > Maybe one way to make this work shine is to provide a 2D plot of speed-accuracy comparison among a series of other works to show the provided trade-off is meaningful. That is, something we could find from pruning/quantization/nas papers a few years ago (although those were usually comparing the inference time with accuracy). For example, Fig1 of Efficientnet.
>
> In response to your suggestion, we have created a 2D plot comparing speed and accuracy among our method and several other relevant works. You can find this plot in Figure 15 on Page 26 in Appendix F.4 of the updated paper. Here are the brief details and data used for this plot:
>
> For the plot, we have considered various efficient methods studies exemplified (but not limited) to the following lines of research:
>
> - use of staleness: which sacrifice the ``freshness of information'' in the training process to accelerate computation or communication in training[1,4]
> - use of quantization: which sacrifices precision to accelerate computation [3]
> - use of estimation: use an estimation method to accelerate the efficiency of the model [2]
>
> We have sampled methods from each category to create a comparison between the relative speed-up and their effect on accuracy. The values of the sampled methods are obtained as follows.
>
> - The value of SAPipe is taken from [1]. The underlying tasks for these values are image classification tasks and language translation tasks.
> - The value of FastGCN is taken from [2] and the underlying task is node classification.
> - PRES(our) is computed by averaging the values from Table. 1 of our paper
>
> The rest of the values are obtained from running the open-source code[6,7,8].
>
> - AdaQP code link [6]: node classification with GCN on the public OGB-product  dataset
> - PipeGCN code link [7]: node classification with GCN on the public OGB-product dataset
> - Sancus code link [8]: node classification with GCN the public OGB-product dataset
>
> The data used for the plot are as follows. The methods with * indicate the values are taken from the original paper.
>
> |  | SAPipe*[1] | FastGCN*[2] | AdaQP[3] | Sancus[4] | PipeGCN[5] | PRES(our) |
> | --- | --- | --- | --- | --- | --- | --- |
> | Average Relative Speed-Up | ~2.3x | ~2x | ~2.5x | ~1.6x | ~2.1x | ~2.6x |
> | Average Drop on Accuracy | ~1.7% | ~1.2% | ~0.9% | ~2.3% | ~1.1% | ~0.7% |
>
> We note that the 2D plot is just a rough comparison, as these methods may not be directly comparable due to different domains and tasks. However, the plot demonstrates that our method provides a reasonable balance between accuracy and speed.
>
> [1]Chen, Yangrui, et al. "SAPipe: Staleness-Aware Pipeline for Data Parallel DNN Training." *Advances in Neural Information Processing Systems* 35 (2022): 17981-17993.
>
> [2] Chen, Jie, Tengfei Ma, and Cao Xiao. "Fastgcn: fast learning with graph convolutional networks via importance sampling." *arXiv preprint arXiv:1801.10247* (2018).
>
> [3] Wan, Borui, Juntao Zhao, and Chuan Wu. "Adaptive Message Quantization and Parallelization for Distributed Full-graph GNN Training." *Proceedings of Machine Learning and Systems* 5 (2023).
>
> [4] Peng, Jingshu, et al. "Sancus: staleness-aware communication-avoiding full-graph decentralized training in large-scale graph neural networks." *Proceedings of the VLDB Endowment* 15.9 (2022): 1937-1950.
>
> [5] Wan, Cheng, et al. "Pipegcn: Efficient full-graph training of graph convolutional networks with pipelined feature communication." *arXiv preprint arXiv:2203.10428* (2022).
>
> [6] https://github.com/raywan-110/AdaQP
>
> [7] https://github.com/GATECH-EIC/PipeGCN/tree/main
>
> [8] https://github.com/chenzhao/light-dist-gnn.

---

> > ### Author Response · Authors · 2023-11-23
> > **Response (2/2)**
> >
> > > I still cannot see the result on page 24. could you check again if the revision has been correctly uploaded
> >
> > We have made sure the ablation study of PRES on the LastFM dataset is correctly uploaded.
> >
> > The problem might arise from multiple-defined labels that caused a mismatch between the reference in the text and the actual figure. We have fixed this problem in the updated version of the paper. Additionally, to enhance the clarity and identification of experiment figures, we have increased the spacing between them.
> >
> > You can now find the LastFM result in Figure 13 on Page 25 (located in the top half of the page). We appreciate your patience and understanding in this matter.
> >
> > > On PipeGCN, could you check if we are looking at the same curve? I believe we should be comparing GCN vs PipeGCN-GF and they both have converged at 250 epochs.
> >
> > We were comparing using the standard PipeGCN.  It is true that PipeGCN-GF would have very similar behaviour as GCN in Fig.4.
> >
> > In our runs of PipeGCN above, we observe that there is almost no accuracy drop with PipeGCN-GF on the Reddit dataset, but there is a slight accuracy drop (~1.1%) with the OGBN-product.
> >
> > > Other experiment results
> >
> > - Extended Training Session
> >
> > In response to your request for an extended training session, we have conducted experiments using TGN on both the WIKI and MOOC datasets on a longer training session (500 epochs). You can find the corresponding plots in Figure 16 on Page 26 in the appendix of the updated paper. Figure 16(a) presents the results of the TGN model with 500 epochs on the WIKI dataset, and Figure 16(b) presents the results of the TGN model with 500 epochs on the MOOC dataset.
> >
> > Figure 16 demonstrates that many of the (minor) discrepancies in accuracy (~0.6%) can be mitigated by or attributed to fluctuations arising from the extended training session.
> >
> >
> > - Combining with Dist-TGL
> >
> > We are afraid integrating Dist-TGL completely with our method is not feasible within the rebuttal period. Dist-TGL is a relatively new library for dynamic (temporal) graph neural networks (the paper was published just last month in SC 23). We do not yet possess a comprehensive understanding of Dist-TGL to the extent required to integrate our method into its modules within the limited rebuttal timeline. We appreciate your understanding in this regard.
> >
> > However, we have made a simple attempt to integrate the mini-batch parallelism of Dist-TGL (creating mini-batches adhering to temporal order) with our method. The following data point is TGN on WIKI dataset with 2 GPUs
> >
> >
> > | Wiki | 2 GPU (Dist-TGL) | 1 GPU |
> > | --- | --- | --- |
> > | TGN (500 Batch Size) | 97.4%, 50 Epochs, 54 Second/Epoch | 97.7%, 50 Epochs, 87 Second/Epoch |
> > | TGN （1200 Batch Size) | 93.4%, 50 Epochs, 19 Second/Epoch | 93.6%, 50 Epochs, 25 Second/Epoch |
> > | TGN-PRES (1200 Batch Size) | 97.6%,  50 Epochs, 20 Second/Epoch | 97.5%, 50 Epochs, 27 Second/Epoch |
> >
> > The table shows that when using the same batch size, the training process can benefit from Dist-TGL. When increasing the batch size, our method can prevent performance degradation.   This suggests that our method is beneficial to Dist-TGL (at least from the perspective of mini-batch parallelism).
> >
> > Thank you once more for the feedback. We hope that our response has addressed your concerns and provided clarifications that further emphasize the significance and contributions of our work.

---

> ### Comment · Reviewer_dnLR · 2023-11-24
> **thanks for the discussion.**
>
> Thanks for the results and further discussion. I checked the results from LastFM.
> I agree that asking for comparison with EDGE and DistTGL was a bit too much considering the timeframe and code availability.
>
> However, I am a bit worried about the 2d plot provided by the authors. As the authors mentioned, they are from different domains, and the value of trade-off is not consistent across domains. Furthermore, gathering data from single dataset might look like cherrypicking, combined with the clipped axes of the plot.
>
> On the other hand, I think having not enough papers to the plot speaks for itself in that this work is timely. Taking the discussions altogether into consideration, I will change my rating to 6. If the paper gets accepted, the 2D plot might be better to be erased from the paper in my opinion, but I would leave that to the decision of the authors.

---

> > ### Author Response · Authors · 2023-11-24
> >
> > Thank you for your thoughtful and constructive feedback on our paper. We truly appreciate your time and effort in reviewing our work.
> >
> > We really appreciate your understanding regarding the comparisons with EDGE and DistTGL.
> >
> > Regarding the 2D plot, we understand your concerns and will carefully consider your suggestion to remove the plot from the paper. Your feedback is important to us, and we will take it into account as we make revisions to our paper.
> >
> > Once again, thank you for your valuable feedback and for changing your rating.

---

### Author Response · Authors · 2023-11-16
**Summary of the updates in the revised paper**

In response to the reviewers' feedback, we have incorporated the following updates into the revised paper:

- Added experimental results utilizing an additional dataset, GDELT, which consists of approximately 17k vertices and 2 million events.
- Provided the missing results for the LASTFM dataset in Appendix F.
- Expanded the discussion in the extended related work section to include recent related works.
- Included additional statistics regarding the absolute running time.

---

### Public Comment · ~Aniq_Ur_Rahman1 · 2024-09-19
**Proofs of Theorem 1 and 2**

I have a few doubts regarding the proofs in Appendix B and C.

### **Proof of Theorem 1**
In eq (13) the last line should be an inequality $ \geq $ instead of $ = $ because $\mathbb{E}[ X^2 ] \geq \mathbb{E}[X]^2 $. Or is there something I am missing?

### **Proof of Theorem 2**
- In eq (16) the loss function $\mathcal{L}$ is defined loosely, i.e., sometimes only with the parameter $\theta$, and sometimes with the batch $\mathcal{B}$.
- Why were $\mu$ and $\sigma^2$ introduced in eq (21)? Since it is an upperbound, and $\mu \in (0,1)$ $^{\star}$, we can multiply any positive term by $\frac{1}{\mu}$, and any negative term by $\mu$, and the inequality will hold. The same can be said for higher powers of $\mu$.
- Also, it is not clear where the line introducing eq (27) comes from.

$^{\star}$ Please correct me if I'm wrong.

---

> ### Public Comment · ~Junwei_Su1 · 2024-09-20
>
> Thank you for taking interest in our paper and reaching out.
>
> Theorem 1:
> We will double-check on the sign to see if there is a typo. It should be noted that switching $=$ with $\geq$ does not change the correctness of the proof because theorem 1 is about the lower bound.
>
> Theorem 2:
> - The loss function with/without $\mathcal{B}$ referred to different things. One is the loss for the complete epoch and the other is for the given batch only. This is defined/described by the main text
> - the $\mu$ is introduced to bound the inner product of gradient from the previous line and $\sigma^2$ is a standard assumption for SGD-type method to bound the variation in gradient.
> - Eq (27) comes from the substitution of the previous line into (26) and with some rearrangement

---

### Meta-Review · Area_Chair_nG9S · 2023-12-13

**Metareview:**

### Summary
This paper explores the challenges in training Memory-based Dynamic Graph Neural Networks (MDGNNs) due to temporal discontinuity during batch processing. It introduces PRES, an approach that employs an iterative prediction-correction scheme and a memory coherence learning objective to alleviate this issue. By addressing the temporal discontinuity problem, PRES enables MDGNNs to be trained with significantly larger temporal batches without sacrificing performance, demonstrating a 4x increase in batch size (3.4x speed-up) during training while maintaining generalization accuracy.

### Decision
The paper is well-written, clear, and easy to follow. The reviewers have raised some concerns related to the paper on issues with the writing and experiments. However, the authors have done a great job of addressing them. Nevertheless, the updates and changes that reviewers have made to the paper did not change the reviewers' score from borderline acceptance. After reading through the paper and reviewing the rebuttal, it is a difficult decision. I think the improvements the authors have made in the efficiency of MDGNNs can enable the practical applications of these models. Although this paper is on the border, I think the improvements the authors demonstrate in this paper might be of interest to the ICLR community.

**Justification For Why Not Higher Score:**

The paper is well-written, and the authors successfully addressed concerns raised by the reviewers. However, it is still not clear how important or the impactful the results achieved in this paper would be.

**Justification For Why Not Lower Score:**

The results are convincing and interesting. The approach proposed in this paper would be beneficial for the broader ICLR community to be aware of.

---

### Decision · Program_Chairs · 2024-01-16

Accept (poster)